



# 1 Stripping back the Modern to reveal Cretaceous climate and temperature gradient

# 2 underneath

Marie Laugié[1], Yannick Donnadieu[1], Jean-Baptise Ladant[2], Mattias Green[3], Laurent Bopp[4,5] and François Raisson[6].
[1]Aix Marseille Univ, CNRS, IRD, INRA, Coll. France, CEREGE, Aix-en-Provence, France
[2]Department of Earth and Environmental Sciences, University of Michigan, Ann Arbor, MI, USA
[3]School of Ocean Sciences, Bangor University, Menai Bridge, UK
[4]Ecole Normale Supérieure (ENS Paris) - Département des Géosciences - France
[5]Laboratoire de Météorologie Dynamique (UMR 8539) (LMD) - Université Pierre et Marie Curie -
Paris 6, Institut national des sciences de l'Univers, École Polytechnique, École des Ponts ParisTech,
Centre National de la Recherche Scientifique : UMR8539, École Normale Supérieure - Paris - France
[6]Total EP – R&D Frontier Exploration - France

## 13 ABSTRACT

During past geological times, the Earth suffered several intervals of global warmth but their driving
factors remain equivocal. A careful appraisal of the main processes involved in those past events is essential to
evaluate how they can inform future climates, and thus to provide decision makers with a clear understanding of
the processes at play in a warmer world. In this context, the greenhouse Earth of the Cretaceous era, specifically
the Cenomanian-Turonian (~94 Ma), is of particular interest, as it corresponds to a thermal maximum. Here we
use the IPSL-CM5A2 Earth System Model to unravel the forcing parameters of the Cenomanian-Turonian
greenhouse climate. We perform six simulations with an incremental change in five major boundary conditions in
order to isolate their respective role on climate change between the Cretaceous and the preindustrial. Starting
with a preindustrial simulation, we implement: (1) the absence of polar ice sheets, (2) the increase in
atmospheric $p$CO$_2$ to 1120 ppm, (3) the change of vegetation and soil parameters, (4) the 1% decrease in the
Cenomanian-Turonian value of the solar constant and (5) the Cenomanian-Turonian paleogeography. Between
the first (preindustrial) simulation and the last (Cretaceous) simulation, the model simulates a global warming of
more than 11°C. Most of this warming is driven by the increase in atmospheric $p$CO$_2$ to 1120 ppm.
Paleogeographic changes represent the second major contributor to the global warming while the reduction in
the solar constant counteracts most of the geographically-driven global warming. We also demonstrate that the
implementation of Cretaceous boundary conditions flattens the temperature gradients compared to the
piControl simulation. Interestingly, we show that paleogeography is the major driver of the flattening in the low-
to mid-latitudes whereas the $p$CO$_2$ rise and polar ice sheet retreat dominate the high-latitudes response.


## 1. INTRODUCTION

The Cretaceous era is of particular interest to understand the drivers of past greenhouse climates because of its prolonged episode of global warmth (O'Brien et al. 2017, Huber et al. 2018), specifically during the thermal maximum of the Cenomanian-Turonian (CT) interval (94 Ma). Proxy-based reconstructions and model simulations of sea-surface temperatures (SST) for the CT reveal that equatorial Atlantic was 4-6° warmer than today (Bice et al., 2006; Norris et al., 2002; Pucéat et al., 2007; Tabor et al., 2016) or even more (6-9° - Forster et al., 2007) during the Oceanic Anoxic Event 2 (OAE2). This is a short and abrupt episode of major climatic, oceanographic and global carbon cycle perturbations occurring at the CT Boundary and superimposed on the long-term global warmth (Jenkyns, 2010). High latitudes were also much warmer than today (Herman and Spicer, 2010; Spicer and Herman, 2010), as was the deep-sea with bottom temperatures reaching up to 20°C during the CT (Friedrich et al., 2012; Huber et al., 2002; Littler et al., 2011). Atmospheric temperatures were also very high as revealed by paleobotanical studies (Herman and Spicer, 1996) with high latitudes temperatures up to 17°C higher than today (Herman and Spicer, 2010) and possibly reaching annual means of 10-12°C in Antarctica (Huber et al., 1999). The steepness of the equator-to-pole gradients is still a matter of debate, in particular because of inconsistencies between data and models as the latter usually predict steeper equator-to-pole gradients (Barron, 1993; Heinemann et al., 2009; Huber et al., 1995; Tabor et al., 2016). Models and data generally agree, however, about a flattened SST gradient relative to today (Huber et al., 1995; Jenkyns et al., 2004; O'Brien et al., 2017; Robinson et al., 2019; Sellwood et al., 1994).

The main factor generally considered as responsible for the Cretaceous global warmth is the higher level of atmospheric $CO_2$ (Barron et al., 1995; Crowley and Berner, 2001; Foster et al., 2017; Royer et al., 2007). Several studies have aimed at reconstructing Cretaceous $p CO_2$ using various techniques, for instance based on the analysis of paleosols $\delta^{13}C$ (Hong and Lee, 2012; Leier et al., 2009; Sandler and Harlavan, 2006), liverworts $\delta^{13}C$ (Fletcher et al., 2006) or phytane $\delta^{13}C$ (Van Bentum et al., 2012; Damsté et al., 2008) or on leaf stomata analysis (Barclay et al., 2010; Mays et al., 2015). Modelling studies have also widely investigated the conundrum of Cretaceous $p CO_2$ question (Barron et al., 1995; Berner, 2006; Bice et al., 2006; Monteiro et al., 2012; Poulsen et al., 2001, 2007) in an attempt to refine the wide range obtained from the data (<900 to >5000 ppm). The typical atmospheric $p CO_2$ concentration resulting from these studies for the CT averages around a long-term value of 1120 ppm (Barron et al., 1995; Bice and Norris, 2003; Royer, 2013; Wang et al., 2014) which is equivalent to four times the preindustrial value (280 ppm = 1 P.A.L) . The $p CO_2$ level is however known to vary on shorter timescales



during this period, in particular during OAE2, which may have been caused by a large increase in atmospheric
pCO2, possibly reaching 2000 ppm or even higher, and which could be attributed to large igneous provinces
volcanic activity (Jenkyns, 2010; Kerr and Kerr, 1998; Turgeon and Creaser, 2008). Proxy records suggest that the
atmospheric $CO_2$ may then have dropped down to 900 ppm after carbon sequestration into organic-rich marine
sediments (Van Bentum et al., 2012).

Paleogeography is also considered as a major driver of climate change through geological times (Crowley

et al., 1986; Goddéris et al., 2014; Gyllenhaal et al., 1991; Lunt et al., 2016). Several processes linked to
paleogeographic changes have been shown to impact the Cretaceous climate. This includes albedo and
evapotranspiration feedbacks from paleovegetation (Otto-bliesner and Upchurch, 1997), seasonality due to
continental break-up or presence of epicontinental seas (Fluteau et al., 2007), atmospheric feedbacks due to
water cycle modification (Donnadieu et al., 2006), Walker and Hadley cells changes after Gondwana break-up
(Ohba and Ueda, 2011) or oceanic circulation changes due to gateways opening (Poulsen et al., 2001, 2003).
Other potential controlling factors include the solar constant, which has been shown to change through time
(Gough et al. 1981) and whose impact on Cretaceous climate evolution was quantified by Lunt et al. (2016), and
changes in the distribution of vegetation, which has been suggested to drive warming, especially in the high-
latitudes with a temperature increase of up to 4°-10°C in polar regions (Brady et al., 1998; Deconto et al., 2000;
Hunter et al., 2013; Otto-bliesner and Upchurch, 1997; Upchurch, 1998).

Despite all these studies, there is no established consensus on the relative importance of these

controlling factors on the CT climate. In particular, the primary driver of Cretaceous climate has suggested to be
either $pCO_2$ or paleogeography. Early studies suggested a negligible role of paleogeography on global climate
compared to the high $CO_2$ concentrations (Barron et al., 1995) whereas others suggested that $CO_2$ was not the
primary control (Veizer et al., 2000), or that the impact of paleogeography on climate was as important as a
doubling of $pCO_2$ (Crowley et al., 1986)s. More recent modeling work has also suggested that paleogeographic
changes could affect global climate (Donnadieu et al., 2006; Fluteau et al., 2007; Poulsen et al., 2003) and the
latest work using coupled climate models are still divided on the climatic impact of paleogeography (Ladant and
Donnadieu, 2016; Lunt et al., 2016; Tabor et al., 2016). For instance, Lunt et al. (2016) support a significant role
of paleogeography at the regional rather than global scale and even show that the global paleogeographic signal
is completely cancelled by an opposite trend due to solar constant changes. Tabor et al. (2016) also support an
important regional climatic impact of paleogeography but argue that $CO_2$ is the main responsible for the Late



Cretaceous climate evolution. In contrast, Ladant and Donnadieu (2016) find a significant impact of
paleogeography on the mean global Late Cretaceous temperatures, roughly comparable to a doubling of
atmospheric $p$CO$_2$. Finally, the role of paleovegetation is also uncertain since some studies show a major role at
high-latitude (Hunter et al., 2013; Upchurch, 1998) whereas more recent work demonstrates limited impact at
high latitudes (<2°C), with a cooling effect at low latitudes, under high $p$CO$_2$ values (Zhou et al., 2012).
In this study, we investigate the forcing parameters of the CT greenhouse climate by using a set of
simulation run with the IPSL-CM5A2 Earth System Model. We performed six simulations, from the preindustrial
to the Cretaceous, and incrementally implement one additional boundary condition change among the following:
(1) the absence of polar ice sheets, (2) the increase in $p$CO$_2$ to 1120 ppm, (3) the change of vegetation and soil
parameters, (4) the 1% reduction in the value of the solar constant and (5) the Cenomanian-Turonian
paleogeography. We particularly focus on the processes driving warming or cooling of the atmospheric surface
temperature under each boundary condition change in order to study the relative importance of each parameter
in the Cenomanian-Turonian to preindustrial climate change. We also investigate how the SST gradient responds
to boundary condition changes in order to understand the evolution of its steepness between the Cretaceous
and the present.

## 2.  MODEL DESCRIPTION & EXPERIMENTAL DESIGN

### 2.1 IPSL-CM5A2 MODEL

IPSL-CM5A2 is an updated version of the IPSL-CM5A-LR earth system model developed at the IPSL (Institut
Pierre-Simon Laplace) as part of the CMIP5 (Dufresne et al., 2013). It is a coupled Earth system model capable of
simulating interactions between atmosphere, ocean and sea ice, and land surface. It also includes the marine
carbon and other key biogeochemical cycles (C, P, N, Fe and Si - See Aumont et al., 2015). IPSL-CM5A has a rich
history of applications, including present-day and future climates (Aumont and Bopp, 2006; Swingedouw et al.,
2017), the IPCC AR5 exercise and CMIP5 project (Dufresne et al., 2013), preindustrial studies (Gastineau et al.,
2013) and paleoclimate (Bopp et al., 2017; Contoux et al., 2015; Kageyama et al., 2013; Sarr et al., 2019; Tan et
al., 2017). IPSL-CM5A has also been used to explore the links between marine productivity and climate (Bopp et
al., 2013; Ladant et al., 2018; Le Mézo et al., 2017), vegetation and climate (Contoux et al., 2013; Woillez et al.,
2014) or topography and climate (Maffre et al., 2018), but also the role of nutrients in the global carbon cycle
(Tagliabue et al., 2010) or the variability of the oceanic circulation and upwelling (Ortega et al., 2015;
Swingedouw et al., 2015). Following technical developments on the components of IPSL-CM5A described in





Sepulchre et al. (2019), the new IPSL-CM5A2 model provides enhanced computing performances allowing long
thousand-years  long integrations required for deep-time paleoclimate applications or long-term future
projections.

22         In details, IPSL-CM5A2 is composed of the LMDz atmospheric model (Hourdin et al., 2013), the ORCHIDEE

land surface and vegetation model (continental hydrological cycle, vegetation, carbon cycle; Krinner et al., 2005)
and the NEMO ocean model (Madec, 2012) including the LIM2 sea-ice model (Fichefet and Maqueda, 1997) and
the PISCES marine biogeochemistry model (Aumont et al., 2015). The OASIS coupler (Valcke et al., 2006) ensures
a good synchronization of the different components and the XIOS input/output parallel library is used to read and
write data. The LMDZ atmospheric component has a horizontal resolution of 96x95, (equivalent to 3.75° in
longitude and 1.875° in latitude) and 39 uneven vertical levels. ORCHIDEE shares the same horizontal resolution
whereas NEMO – the ocean component – has 31 uneven vertical levels (from 10 meters at the surface to 500
meters at the bottom), and a horizontal resolution of approximately 2°, enhanced to up to 0.5° in latitude in the
tropics. NEMO uses the ORCA2.3 tripolar grid to overcome the North Pole singularity (Madec and Imbard, 1996).
IPSL-CM5A2 and its performances in simulating preindustrial and modern climates are fully described in
Sepulchre et al. (2019).
2.2 EXPERIMENTAL DESIGN

35         Six simulations have been performed for this study: one preindustrial control simulation named *piControl*

and five simulations for which boundary conditions were changed incrementally to progressively reconstruct the
Cretaceous conditions (Table 1): (1) *1X-NOICE* with polar ice caps retreat, *4X-NOICE* (pCO$_2$ at 1120 ppm), *4X-*
*NOICE-PFT-SOIL* (implementation of idealized Plant Functional Types (PFTs) and mean parameters for soil), *4X-*
*NOICE-PFT-SOIL-SOLAR* (reduction of the solar constant) and *4X-CRETACEOUS* (CT paleogeography). The piControl
simulation was run for 1800 years and the five others for 2000 years in order to reach the equilibrium state
(Fig.1).

42         The piControl and 1X-NOICE simulations are initialized with Atmospheric Model Intercomparison Project

(AMIP) conditions that were constrained by realistic sea surface temperature (SST) and sea ice from 1979 to near
present (Gates et al., 1999). Modern boundary conditions of NEMO include forcings of the dissipation associated
with internal wave energy for the M2 and K1 tidal components (de Lavergne et al., 2019). The parameterization
follows Simmons et al. (2004) with refinements in the modern Indonesian Through Flow (ITF) region according to
Koch-Larrouy et al. (2007). As most evidence suggests the absence of permanent polar ice sheets during the CT



(Huber et al., 2018; Ladant and Donnadieu, 2016; MacLeod et al., 2013), we isostatically remove polar ice sheets
in the 1X-NOICE simulation and replace them with brown bare soil. In the 4X-NOICE simulation, polar ice caps are
also removed and the $pCO_{2\,is}$ fixed to 1120 ppm (four times the "Preindustrial Atmospheric Level" [P.A.L] ), a
value reasonably close to the mean suggested by a recent compilation of Cretaceous $pCO_2$ reconstructions
(Wang et al., 2014). In an attempt to reach the equilibrium state faster, the initial conditions of simulations with
4x PAL are taken from idealized conditions (higher SST and no sea ice) similar to those described in Lunt et al.,
2017. We keep the 13 PFTs of ORCHIDEE but their distribution is reassigned along latitudinal bands, based on a
rough comparison with the preindustrial distribution of vegetation, in order to obtain a theoretical latitudinal
distribution usable for any geological period. The list of PFTs and associated latitudinal distribution and fractions
are described in the Supplementary Table 1. The soil parameters, i.e., the mean color and texture (rugosity), are
calculated from preindustrial maps (Wilson and Henderson-sellers, 2003; Zobler, 1999) and assigned to the
whole world. The impact of these idealized PFTs and mean parameters is discussed in the results. The 4X-NOICE-
PFT-SOIL-SOLAR is initialized from the same conditions as 4X-NOICE-PFT-SOIL except that the solar constant is
reduced to its Cretaceous value (Gough, 1981). We use here the value of 1351.36 W/m$^2$ (98.9% of the Modern
solar luminosity, calculated for an age of 90 My). Finally, the 4X-CRETACEOUS simulation incorporates the
previous modifications plus the implementation of the CT paleogeography. The land-sea configuration used here
is the one proposed by Sewall (2007) for the CT, in which we have implemented the bathymetry from Müller
(2008) (Fig. 2). These bathymetric changes are done in order to represent deep oceanic topographic features,
such as ridges, that are absent from the Sewall paleogeographic configuration. In this simulation, the mean soil
color and rugosity as well as the theoretical latitudinal PFTs distribution are adapted to the new land-sea mask
and the river routing is recalculated from the new topography. To create a cenomanian-turonian dissipation
forcing, we used a M2 tidal field calculated using the Oregon State University Tidal Inversion System (OTIS, Egbert
et al., 2004; Green and Huber, 2013). The M2 field is computed using our cenomanian-turonian bathymetry and
an ocean stratification taken from an equilibrated cenomanian-turonian simulation realized with the IPSLCM5A2
with no M2 field. In the absence of any estimation for the Cretaceous, we prescribe the K1 tidal field to 0. In
addition, the parameterization of Koch-Larrouy et al. (2007) is not used here because the ITF does not exist in the
Cretaceous.



*Table 1 : Description of the simulations. The parameters in bold indicate the specific change for the corresponding simulation. Simulations are run for 2000 years, except piControl which is run*
*for 1000 years.*

| Simulation | piControl | 1X-NOICE | 4X-NOICE | 4X-NOICE-PFT-SOIL | 4X-NOICE-PFT-SOIL-SOLAR | 4X-CRETACEOUS |
|---|---|---|---|---|---|---|
| Polar Caps | Yes | **No** | No | No | No | No |
| CO₂ (ppm) | 280 | 280 | **1120** | 1120 | 1120 | 1120 |
| Vegetation | IPCC (1850) | IPCC (1850) + Bare soil instead of polar caps | IPCC (1850) + Bare soil instead of polar caps | **Theoretical latitudinal PFTs** | Theoretical latitudinal PFTs | Theoretical latitudinal PFTs |
| Soil Color/Texture | IPCC (1850) | IPCC (1850) + Brown soil instead of polar caps | IPCC (1850) + Brown soil instead of polar caps | **Uniform mean value** | Uniform mean value | Uniform mean value |
| Solar constant (W/m²) | 1365.6537 | 1365.6537 | 1365.6537 | 1365.6537 | **1353.36** | 1353.36 |
| Geographic configuration | Modern | Modern | Modern | Modern | Modern | **Cretaceous 90 Ma (Sewall 2007 + Müller 2008)** |



# 3. RESULTS

The simulated changes between the preindustrial simulation (piControl) and the Cretaceous simulation (4X-CRETACEOUS) can be decomposed into five components: (1) Polar cap retreat ($\Delta$ice), (2) $p$CO$_2$ ($\Delta$CO$_2$), (3) PFT and Soil parameters ($\Delta$PFT-SOIL), (4) Solar constant ($\Delta$solar) and (5) Paleogeography ($\Delta$paleo). Each contribution on global climate change can be calculated by a linear factorization (Broccoli and Manabe, 1987; Von Deimling et al., 2006), which simply corresponds to the anomaly between two consecutive simulations. The results presented in the following are averages calculated over the last 100 simulated years (out of 2000).

## 3.1 GLOBAL

The progressive change of parameters made to reconstruct Cretaceous climate induce a general global warming (Table 2, Fig. 3). The annual global atmosphere temperature at 2 meters above the surface (T2M) rises from 13.25°C to 24.35 °C, which represents an increase of 84%. The majority of this warming, 9°C or a contribution of 61% of the cumulative absolute temperature change between the preindustrial and the CT the signal, comes from increasing the pCO$_2$ to 4 P.A.L. The paleogeography also represents a major contributor to the warming, with an increase in T2M of 2.6°C (18%). The impact of the solar constant decrease contributes to 12% of change (1.8°C), but with an opposite effect of cooling. Finally, changes in the soil parameters and PFTs as well as the retreat of polar caps have a minor impact on the global T2M (6% and 3% change, respectively).

The temperature changes have a different geographic response (Fig. 4) depending on the changed parameter, ranging from a global and uniform cooling ($\Delta$solar – Fig 4e) to a global warming ($\Delta p$CO$_2$ – Fig 4c), via contrasted regional responses ($\Delta$ice or $\Delta$paleo – Fig 4b and 4f). In the next section, we describe the main patterns of change and the main feedbacks arising.

| | | piControl | 1X-NOICE | 4X-NOICE | 4X-NOICE-PFT-SOIL | 4X-NOICE-PFT-SOIL-SOLAR | 4X-CRETACEOUS |
|---|---|---|---|---|---|---|---|
| **T2M (°C)** | Global Anomaly | +11.1 → +84% | | | | | |
| | Results | 13.25 | 13.75 | 22.75 | 23.55 | 21.75 | 24.35 |
| **Planetary Albedo (%)** | Global Anomaly | -5.94 → -18% | | | | | |
| | Results | 33.07 | 32.61 | 28.79 | 28.27 | 28.66 | 27.13 |





| Surface Albedo (%) | Global Anomaly | -5.19 ➔ -26% | | | | | |
|---|---|---|---|---|---|---|---|
| | Results | 20.13 | 19.02 | 16.56 | 15.46 | 15.35 | 14.94 |
| Emissivity (%) | Global Anomaly | -4.97 ➔ -8% | | | | | |
| | Results | 62.01 | 61.7 | 57.51 | 57.14 | 57.77 | 57.04 |

*Table 2 : Simulations results (Global annual mean over last 10 years of simulation) and calculated anomaly between*
*4XCRETACEOUS and piControl simulations.*

## 3.2 The major contributor to global warming - ΔCO$_2$

207   The fourfold increase in $p$CO$_2$ leads to a global warming of 9°C (Table 3, Fig. 3) between 1X-
NOICE-and 4X-NOICE simulations. The whole surface is warmer with an amplification located over the
Arctic and Austral oceans and which is generally larger over continents than over oceans (Fig 4c). The
warming is due to a general decrease of planetary albedo and of the atmosphere's emissivity. The
decrease in atmosphere's emissivity is directly driven by the increase of CO$_2$, and thus greenhouse
trapping in the atmosphere, but it is also amplified by an increase in high-altitude cloudiness over the
Antarctic continent (Fig 5a,b). The decrease in planetary albedo is due to (1) a decrease of sea ice and
snow (especially over Northern hemisphere continents and along the coasts of Antarctica) and thus of
surface albedo, which explain the warming amplification over polar oceans, and (2) a decrease in low-
altitude cloudiness (except over the Arctic - Fig 5a,b). The decrease in low-altitude cloudiness is linked
to a decrease of relative humidity in areas of formation of low clouds (outside of the tropics), despite a
general increase of evaporation and specific humidity. The relative humidity decrease can be driven by
the temperature rise associated with enhanced greenhouse trapping, allowing the atmosphere to hold
more moisture. Over the Arctic, an increase in low-altitude cloudiness is simulated despite the
decrease in relative humidity, and can be described by a strong sea level pressure decrease allowing
more air to rise and more clouds to form. This sea level pressure decrease is possibly a feedback
driven by the sea ice melting and associated higher temperatures. Here, the increase in low-altitude
cloudiness acts as a negative feedback and attenuates the warming induced by the surface albedo
decrease. Nevertheless, its impact is minor given the strong atmospheric temperature warming
observed over the Arctic (Fig 4c).
227   The contrast in the response of the atmosphere over continents and oceans is due to the
impact of the evapo-transpiration feedback. The warming drives an increase of evaporation, which
acts as a negative feedback and moderates the warming by consuming more latent heat at the ocean
surface. In contrast, high temperatures tend to inhibit vegetation development over continents, which



acts as positive feedback and enhances the warming due to reduced transpiration and reduced latent
heat consumption.

**3.3 Boundary conditions with the smallest global impacts – Δice, ΔPFT-SOIL, Δsolar**
The polar cap retreat, in 1X-NOICE simulation, leads to a weak global warming of 0.5°C but a
strong regional warming observed over areas previously covered by ice caps (Antarctica and
Greenland – Fig 4a,b). This is due to a combination of a decrease in elevation and of surface albedo,
which is directly linked to the removal of polar ice sheets. Unexpected cooling is also simulated in
specific areas, such as the margins of the Arctic Ocean and the southwestern Pacific. These contrasted
climatic responses to the impact of ice sheets on sea surface temperatures have been observed in
previous modeling studies but their origin is still unclear (Goldner et al., 2014; Kennedy et al., 2015;
Knorr and Lohmann, 2014).
The change in soil parameters and the implementation of theoretical zonal PFTs, in simulation
4X-NOICE-PFT-SOIL, drive a warming of 0.8 °C. This warming is essentially located above arid areas,
such as the Sahara, Australia, or the Middle-East, and polar latitudes (Antarctica/Greenland) (Fig 4d).
The warming above arid areas is mostly caused by the implementation of a mean uniform soil color,
which drives a surface albedo decrease over deserts that normally have a lighter color. The warming at
high latitudes is linked to the vegetation change: the bare soil, that characterizes the continental
regions previously covered with ice, is replaced by boreal vegetation, which drives a surface albedo
decrease. The presence of vegetation at such high latitudes is consistent with high latitude
paleobotanical data ant temperature records during the Cretaceous (Herman and Spicer, 2010; Otto-
bliesner and Upchurch, 1997; Spicer and Herman, 2010).
Finally, the change in solar constant from 1365 W/m$^2$ to 1353 W/m$^2$ directly drives a cooling
of 1.8 °C, rather evenly distributed over the Earth (Fig 4e), see Gough (1981).

**3.4 The most complete response - Δpaleogeography**
The paleogeographic change drives a global warming of 2.6 °C. This is seen all year round in
the Southern Hemisphere, while the Northern Hemisphere experiences a warming during winter and a
cooling during summer (Fig 6). These temperature changes are linked to a general decrease of
planetary albedo and/or emissivity, although the Northern Hemisphere experiences an increased
albedo, due to the increase in low-altitude cloudiness. This trend is compensated by a strong
atmosphere emissivity decrease during winter but not during summer, which leads to the seasonal
pattern of cooling and warming.



The albedo and emissivity changes are linked to atmospheric and oceanic circulation
modifications driven by major features of the new paleogeography (Fig 2):

(1)  Equatorial oceanic gateway opening (Panama/Tethys)

(2)  Polar gateway closure (Drake/Tasman)

(3)  Increase of oceanic area in the North Hemisphere (Fig 2)

(4)  Decrease of oceanic area in the South Hemisphere (Fig 2)


The opening of equatorial gateway creates a zonal connection between the Pacific, Atlantic
and Indian oceans via the Tethys. This connection allows a strong circumglobal equatorial current to
form under the influence of stronger easterly winds because of the absence of continental barriers in
the CT configuration (Fig 7a-b). These winds and currents drive an intensification of upwelling along
the equator and of the surface meridional currents (Fig8a-b), and therefore an increased poleward
ocean heat transport (Fig 8c) (Enderton and Marshall, 2008; Hotinski and Toggweiler, 2003). A similar
line of reasoning can be used in the Southern Hemisphere to explain the increase of southward ocean
heat transport observed between 40° and 60°S (Fig 8c). The modern Antarctic Circumpolar Current
(ACC) does not exist during the Cretaceous because of the closed Drake and Tasman gateways (Fig 7c-
d), and the ocean heat transport is thus enhanced. An altered circulation emerges in the Southern
Ocean, showing a more gyre-like circulation, which allows and increased polar heat transport. The
increased oceanic heat transport is associated with a meridional expansion of high sea-surface
temperatures leading to an intensification of evaporation between the tropics and a shift of the
ascending branches of the Hadley cells a few degrees off the Equator towards the subtropics. The
combination of these two processes results in an increased injection of moisture into the upper
atmosphere and thus in high-altitude cloudiness increase and spreading towards the tropics, leading
to an important greenhouse effect. This process acts as the main factor of intertropical warming
(Herweijer et al., 2005; Levine and Schneider, 2010; Rose and Ferreira, 2013).
The atmosphere response to paleogeographic change in the mid- and high-latitudes is
different in the Southern and Northern Hemispheres as the oceanic areas decrease or increase
between the CT configuration and the modern. In the Southern Hemisphere, the reduced ocean
surface area (Fig 2) limits evaporation and moisture injection into the atmosphere, which in turn leads
to a decrease in relative humidity and low-altitude cloudiness (Supplementary Fig 1) and an associated
year-round warming due to a reduced planetary albedo. In the Northern Hemisphere, the oceanic
area increases (Fig 2) and results in a strong increase of evaporation and moisture injection into the
atmosphere. Low-altitude cloudiness and hence the albedo, both increase and lead to the cooling
during the summer as discussed above (Fig 6). During winter, on the other hand, an increase of high-
altitude cloudiness leads to an enhanced greenhouse effect and counteracts the larger albedo. This



high-altitude cloudiness increase is due to the extratropical increase in OHT (Fig. 8) which enhances
mid-latitude convection and moist air injection into the upper troposphere which spreads towards the
pole (see also Rose and Ferreira, 2013). Also, the increased continental fraction of the Cretaceous
paleogeography leads to a decreased continentality (Donnadieu et al. 2006) because of the thermal
inertia of the oceans that is different to that of continents.

## 3.5 Temperature Gradients
### 3.5.1 Ocean
The mean annual global SST increases of 9.8°C, from 17.9°C to 27.7 °C across the simulations.
This warming is slightly weaker than the mean annual global atmospheric temperature at 2m
discussed above, and most likely occurs because of evaporation processes due to the weaker
atmospheric warming above oceans compared to that above continents. As for atmospheric
temperatures, $pCO_2$ appears as the major controlling parameter of the ocean warming (49% of the
absolute temperature change), followed by paleogeography (30%) and solar constant (16%), although
the latter again drives cooling rather than warming. PFT and soil parameter changes and polar ice cap
retreat instead have a minor impact at the global scale (4% and 0% respectively). It is interesting to
note the increased contribution of paleogeography in the simulated sea surface warming compared to
the atmospheric warming, which is probably driven by the major changes simulated in the surface
circulation (Fig. 7).
The piControl simulation show an average tropical SST average of ~ 26°C (calculated as the
zonal average between 30°S and 30°N) and of ~ -1.5°C at the poles (beyond 70° N - Fig 9a). The
meridional temperature gradients, calculated as the linear temperature change per 1° of latitude
between 30° and 80°, are of 0.45°C/°latitude and 0.44°C/°latitude for the Northern and Southern
Hemispheres, respectively. The Cretaceous simulation yields SST averages of ~ 33.3°C in the tropics
and of ~5°C and 10°C in the Arctic and Southern Ocean respectively. Associated Cretaceous meridional
gradients are of 0.45°C/°latitude and 0.39°C/°latitude for the Northern and Southern Hemispheres,
respectively. The progressive flattening of the SST gradient can be explained by superimposing the
zonal mean temperatures of the different simulation and by adjusting them at the Equator (Fig 9b).
Two major observations can be made from these results. First, paleogeography has a strong impact on
the low-latitudes SST gradient because it widens the latitudinal band of relatively homogeneous warm
tropical SST). As explained before (See Results – Δpaleogeography), this is due to the opening of
equatorial gateways.  Second, the SST gradient beyond 40° of latitude is flattened in two steps with
paleogeography being the major contributor followed by atmospheric $pCO_2$ increase.



### 3.5.2 Atmosphere


In the piControl simulation, tropical atmospheric temperatures are ~ 23.6°C whereas polar

temperatures (calculated as the zonal average between 80° and 90° of latitude) in the Northern and
Southern Hemisphere are around -16.8°C and -37°C respectively. The northern meridional
temperature gradient is 0.69°C/°latitude while the southern latitudinal temperature gradient is
1.07°C/°latitude (Fig 9c). In the Cretaceous simulation, the tropical atmospheric temperatures are ~
32.3°C and polar temperatures ~ 3.4°C in the Northern Hemisphere and ~ - 0.5°C in the Southern
Hemisphere. This yields latitudinal temperature gradients of 0.49°C/°latitude and 0.54°C/°latitude,
respectively. As for the SST gradients, we normalized the curves so that temperatures at the Equator
are equal for each simulation (Fig 9d). This shows that the different mechanisms responsible for the
flattening of the gradients are different for each hemisphere. In the south, at high-latitudes, three
parameters contribute to reducing the equator-to-pole temperature gradient in the following order of
importance: retreat of polar ice caps is the most important, paleogeography and atmospheric $pCO_2$
increase. In contrast, in the Northern Hemisphere, only the rise in $pCO_2$ contributes to reducing the
steepness of the temperature gradient. Furthermore, in the Southern Hemisphere low- to mid-
latitudes, only the paleogeography acts to reduce the steepness of the temperature gradient, whereas
in the northern hemisphere low- to mid-latitudes, both paleogeography and atmospheric $pCO_2$
contribute to this decrease with a similar magnitude.

The role of the polar ice sheet retreat on gradient flattening is considerable but only locally

because of the geographic restriction of the ice sheets. We have seen that the removal of polar ice
sheets only warms areas initially covered by ice whereas equatorial temperatures remain unchanged
Fig. 4b). This explains the regionally flattened gradient in the Southern Hemisphere whereas the
smaller size of the Greenland ice sheet only marginally affects the northern gradient. In contrast, the
increase in $pCO_2$ exerts a more global impact that is amplified at high latitudes because of the sea ice
decrease. Finally, the contribution of paleogeography to the gradient flattening is observed almost at
all latitudes but is clearer in the Southern Hemisphere. In the tropics the impact of paleogeography on
the latitudinal atmospheric gradient is linked to the surface oceanic circulation changes described
previously. In the mid- to high-latitudes the increased continental areas in the Cretaceous Southern
Hemisphere drive the reduction in the steepness of the atmospheric gradient because of the
enhanced warming of continental areas. In the Northern Hemisphere, the increase in oceanic areas
instead tends to obscure the flattening of the atmospheric gradient.




## 4. DISCUSSION

### 4.1 ABOUT THE CENOMANIAN-TURONIAN CLIMATE

The results predicted by our CT simulation were compared to the reconstructed atmospheric and oceanic paleotemperatures from proxy data (Fig 10a,b). The SST data compilation is essentially based on that of Tabor (2016) and includes several proxies such as TEX86, $\delta^{18}O$ of fish teeth, foraminifera and shells, and crocodilian fossil evidence. Atmospheric temperature data are obtained from paleobotanical and paleosoil studies (see supplementary data for the complete database and references). Cretaceous equatorial and tropical SST have long been believed to be similar or even lower than those of today (Crowley and Zachos, 1999; Huber et al., 2002; Sellwood et al., 1994), thus feeding the problem of "tropical overheating" systematically observed in General Circulation Model simulations (Barron et al., 1995; Bush et al., 1997; Poulsen et al., 1998). This incongruence was based on the relatively low tropical temperatures reconstructed from foraminifera (25-30°C, Fig. 9a) but it has later been suggested that these were underestimated (Pearson et al., 2001; Pucéat et al., 2007). The latest data compilations including temperature reconstructions from other proxies (TEX86, $\delta^{18}O$ from shells or fish tooth) have provided support for high tropical SST in the Cenomanian-Turonian (O'Brien et al., 2017; Tabor et al., 2016) and our tropical SST are mostly consistent with existing paleotemperature reconstructions (Fig. 9a). In contrast to the numerous Cenomanian-Turonian SST records, there are, to our knowledge, no atmospheric temperature reconstructions available for tropical latitudes.

In the mid-latitudes (30-60°) the proxy records show a wide range of SST, ranging from 10°C to more than 30°C. We observe that this trend can be reproduced in our simulation when considering the local monthly maximum and minimum temperatures (grey shaded areas, Fig 10a), suggesting a reasonable model-data agreement. Simulated atmospheric temperatures for these latitudes in the Southern hemisphere also show reasonable agreement, whereas the Northern Hemisphere mean zonal temperatures in our model are slightly warmer than that inferred from proxies (Fig 10b).

There are unfortunately only a few high-latitudes SST data points available, which renders the model-data comparison difficult. This is further aggravated by both the proxy-based and simulated SST presenting a large range for a given latitude. In the Northern Hemisphere, temperatures inferred from the presence of crocodilian fossils (Vandermark et al., 2007) in the northern Labrador Sea (~70° of latitude, not represented in our paleogeography) are around 14°C for the annual mean and 5°C for the coldest month. In comparison, the simulated temperatures at the same latitude in the adjacent Western Interior Sea are very similar (13.5 °C for the annual mean and 7.9 °C for the coldest month). In the Southern Hemisphere, the mean annual SST calculated from foraminifera at sites DSDP 511 and 258 (Huber et al., 2018); 55° of latitude in our paleogeography) are between 25° and 30°C whereas





the simulated annual SSTs average at 14.2°C for this latitude. However, the simulated annual SSTs can
reach 19.4°C locally, with a monthly maximum up to 28°C, especially around the location of site DSDP
258. We speculate that a seasonal bias in the foraminiferal record may represent a possible cause for
this difference, as may local deviations of the regional seawater $\delta^{18}O$ from the globally assumed -1‰
value. The same trend is observed for atmospheric temperatures with data indicating higher
temperatures than the model at high latitudes for both southern and northern hemispheres.
However, we observe the same underestimate of simulated high-latitudes atmospheric temperatures
compared to observations in both hemispheres, which could indicate a systematic cool bias of the
simulated temperatures.
The simulated northern latitudinal SST gradient of (~0.45°C/°latitude) is in good agreement
with those obtained by geological data for the northern hemisphere (~0.42°C/°latitude) whereas the
simulated southern latitudinal gradient is significantly higher (~0.39°C/°latitude vs ~0.3°C/°latitude)
(Fig 11). This overestimate of the latitudinal gradient is also true for the atmosphere as data-inferred
gradients are much lower (North=0.2°C/°latitude, South=0.18°C/°latitude) than that of the simulation
(North=0.49°C/°latitude, South=0.55°C/°latitude), although the paucity of Cenomanian-Turonian
continental temperatures proxy data is likely to significantly bias this comparison.
In the following, we compare our simulated gradients to those obtained in previous deep time
modelling studies using recent earth system models. Because such modeling studies focusing on the
Cenomanian-Turonian are limited, we include simulations of the Early Eocene (~ 55 Ma), which is
another interval of global climatic warmth (Lunt et al., 2012a, 2017) (Fig. 11). The simulated SST
latitudinal gradients range from 0.32°C/°latitude to 0.55°C/°latitude (Lunt et al., 2012; Tabor et al.,
2016; Zhu et al., 2019; Fig. 11) and the atmospheric latitudinal gradients from 0.33°C/°latitude to
0.78°C/°latitude (Huber and Caballero, 2011; Lunt et al., 2012; Niezgodzki et al., 2017; Upchurch et al.,
2015; Zhu et al., 2019; See Fig. 11), with the lowest latitudinal gradients being obtained for the highest
$p$CO$_2$ values. The IPSL-CM5A2 is well within the range of other models, which almost systematically
simulate larger gradients than those obtained from data (Fig. 11, see also Huber, 2012). Reasons
behind this incongruence are debated (Huber, 2012) but it highlights the need to get more data and to
challenge the behavior of complex earth system models, in particular in the high latitudes. Studies
have demonstrated that models are able to simulate lower latitudinal temperature gradients under
specific conditions such as anomalously high CO$_2$ concentrations (Huber and Caballero, 2011),
modified cloud properties and radiative parameterizations (Upchurch et al., 2015; Zhu et al., 2019) or
lower paleo elevations and/or more extensive wetlands (Hay et al., 2019). Finally, from a proxy
perspective, it was suggested that a sampling bias could exist, with a better record of temperatures
during the warm season at high latitudes and during the cold season in low latitudes (Huber, 2012).
Such possible biases would help reduce the model-data discrepancy, in particular for atmospheric





temperatures (Fig 10b), as high-latitude reconstructed temperatures are more consistent with
simulated summer temperatures whereas the consistency is better with simulated winter
temperatures in the mid- to low-latitudes, but more work is required to unambiguously demonstrate
the existence of these biases.

4.2 CRETACEOUS CLIMATE CONTROLLING FACTORS

The earliest estimates of the temperature change under a doubling of the atmospheric $p$CO$_2$

predicted a 1.5 to 4.5°C temperature increase, with the most likely scenario at 2.5°C of increase
(Barron et al., 1995; IPCC, 2014; Sellers et al., 1996). Our modelling study predicts an atmospheric
warming of 11.1°C for the Cretaceous. The signal includes 9°C due to the fourfold increase of $p$CO$_2$ or
a 4.5°C increase for a doubling of $p$CO$_2$ (assuming that the response is linear), which agrees with the
high end of the investigations mentioned above. Whilst large, latest generation of earth system
models also show an increasingly higher climate sensitivity to increased CO$_2$ (Golaz et al., 2019;
Hutchinson et al., 2018; Niezgodzki et al., 2017), suggesting that the sensitivity could have been
underestimated in earlier studies. For example, the recent study of Zhu (2019), using an up-to-date
parametrization of cloud microphysics in the CESM1.2 model, proposes an Eocene Climate Sensitivity
of 6.6°C for a doubling of CO$_2$ from 3 to 6 PAL.

$p$CO$_2$ has been shown here to be the main controlling factor for the atmospheric global

warming,  whereas the effects of the paleogeography (warming) and reduced solar constant (cooling)
nearly cancel each other out (see also Lunt et al., 2016). These results agree with previous studies
suggesting that $p$CO$_2$ is the main factor controlling the climate (Barron et al., 1995; Crowley and
Berner, 2001; Foster et al., 2017; Royer et al., 2007). However, we also demonstrate that the
paleogeography plays a major role in the latitudinal distribution of temperatures and impacts oceanic
temperatures (with a similar magnitude as a doubling of $p$CO$_2$), thus confirming that it is also a critical
driver of the Earth's climate (Donnadieu et al., 2006; Fluteau et al., 2007; Lunt et al., 2016; Poulsen et
al., 2003). This large effect on climate by the continental configuration has not been reported for
paleogeographic configurations more similar to each other, e.g., the Maastrichtian and Cenomanian
(Tabor et al., 2016). This is because the main features influencing climate in our study (i.e. the
configuration of equatorial and polar zonal connections and the land/sea repartition) do not change a
lot between the two geological periods investigated by Tabor et al. (2016). Paleogeography is thus a
first-order controller of climate on long scales.

It has been suggested that high latitude warming, and an associated reduced meridional SST

gradient, was amplified in deep time simulations by rising CO$_2$ via cloud and vegetation
feedbacks (Deconto et al., 2000; Otto-bliesner and Upchurch, 1997) or by increasing ocean heat



transport (Barron et al., 1995; Brady et al., 1998; Schmidt and Mysak, 1996), in particular when
changing the paleogeography (Hotinski and Toggweiler, 2003). Our study confirms that the
paleogeography is the primary control on the steepness of the oceanic meridional temperature
gradient. It is also the only process controlling both the atmosphere and ocean temperature gradients
in the tropics. It also has a greater impact than atmospheric CO2 on reducing the atmospheric
temperature gradient at high latitudes in the Southern Hemisphere between the Cretaceous and the
preindustrial, although the major driver here is the retreat of the polar ice sheets. The increase in
$p\text{CO}_2$ appears as the second most important parameter for modifying the SST gradient at high
latitudes and it is the main controller of the reduced atmospheric gradient in the North Hemisphere
due to low clouds albedo feedback. The effect of paleovegetation on the reduced temperature
gradient is not present at high latitudes in our simulations, in contrast to tha warming, up to 4-7°C,
reported elsewhere (Deconto et al., 2000; Otto-bliesner and Upchurch, 1997; Upchurch, 1998). Our
results thus support the limited influence of vegetation in the Cretaceous high-latitudes warmth (Zhou
et al., 2012). However, our modeling setup prescribed boreal vegetation at latitudes higher than 50°
("Boreal broad-leaved summergreen" and "Boreal needleleaf summergreen" PFTs) whereas it was
suggested that evergreen forests could possibly develop beyond 60° of latitude (Hay et al., 2019;
Sewall et al., 2007) and that temperate forests could extend up to 60° of latitude (Otto-bliesner and
Upchurch, 1997). Based on our results, we cannot exclude that this kind of high latitude vegetation
can give more weight to the role of paleovegetation in reducing the temperature gradient.

## 5. CONCLUSIONS

To quantify the impact of major climate forcings on the Cretaceous climate, we performed a
series of 6 simulations using the IPSL-CM5A2 earth system model in which we incrementally
implement changes in boundary conditions on a pre-industrial simulation to obtain in the end a
simulation of the Cenomanian-Turonian stage of the Cretaceous. This study confirms the primary
control exerted by atmospheric $p\text{CO}_2$ on atmospheric temperatures, with a contribution of 61% to the
total absolute global warming. At the global scale, paleogeographic and solar constant changes have
opposite effects, canceling each other, while polar ice cap retreat and vegetation and soil parameter
changes have only minor impact. Atmospheric $p\text{CO}_2$ still explains the majority of the global SST
warming (49%) but the amount of change explained by paleogeography increases compared to the
atmospheric temperature change and thus represents a major contribution (30%). The study of
temperature gradients reveals that the reduction of the meridional SST gradients between the
preindustrial and the Cretaceous is mainly due to the paleogeographic changes and to a lesser extent
to the increase of $p\text{CO}_2$. The atmospheric gradient response is more complex because its flattening is



controlled by several factors including paleogeography, pCO2 and polar ice cap retreat, with different
answers for the Southern and Northern hemispheres. While predicted oceanic and atmospheric
temperatures show a good agreement with data in the low and mid latitudes, predicted temperatures
in the high latitudes are colder than paleotemperatures reconstructed from proxies, which leads to
steeper equator-to-pole gradients in the model than that calculated from proxies. This mismatch often
observed in data-model comparison studies has been reduced in the last decades and could be further
resolved by considering possible sampling/seasonal biases in the proxies and by continuously
improving the model physics and parameterization. Such modelling efforts would probably even more
increase the equilibrium climate sensitivity, which is revised upwards in the latest modelling studies.

## DATA AVAILABILITY

Data that support the results of this study are available on request to the authors.

## AUTHOR CONTRIBUTION

M.L performed and analyzed the numerical simulations, in close cooperation with Y.D and J.B.L, and
led the writing. M.G run the OTIS model to provide the cenonamian-turonian M2 coefficient. All
authors discussed the results and analyses presented in the final version of the manuscript.

## COMPETING INTERESTS

The authors declare that they do not have competing interests.

## AKNOWLEDGMENTS

We express our thanks to Total E&P for funding the project and granting permission to publish. We
thank the CEA/CCRT for providing access to the HPC resources of TGCC under the allocation 2018-
GEN2212 made by GENCI. J.A.M.G receives funding from the Natural Environmental Research Council
(grant NE/S009566/1, MATCH). We acknowledge use of the Ferret (ferret.pmel.noaa.gov/Ferret/)
program for analysis and graphics in this paper.




FIGURES

*Figure 1: Time series for oceanic temperatures. (a) Sea-surface temperature and (b) deep-ocean (2500 m) temperature. The piControl and 1X-NOICE simulations are perfectly equilibrated. The 4X simulations still have a small linear drift, around 0.1°C/century or less : 0.07, 0.08, 0.05 and 0.01°C/century during the last 500 yrs for SST of 4X-NOICE, 4X-NOICE-PFT-SOIL, 4X-NOICE-PFT-SOIL-SOLAR and 4X-CRETACEOUS respectively; 0.11, 0.08, 0.07 and 0.06°C/century during the last 500 yrs, for deep-ocean of 4X-NOICE, 4X-NOICE-PFT-SOIL, 4X-NOICE-PFT-SOIL-SOLAR and 4X-CRETACEOUS respectively).*

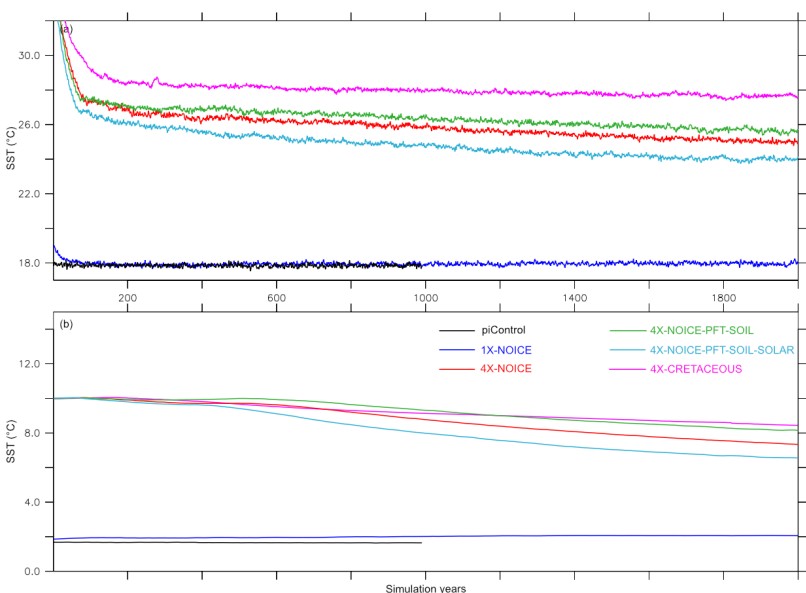



*Figure 2: Modern and Cenomanian-Turonian geographic configurations used for the piControl and 4X-CRETACEOUS simulations respectively, and meridional oceanic area anomaly between Cretaceous paleogeography and Modern geography.*

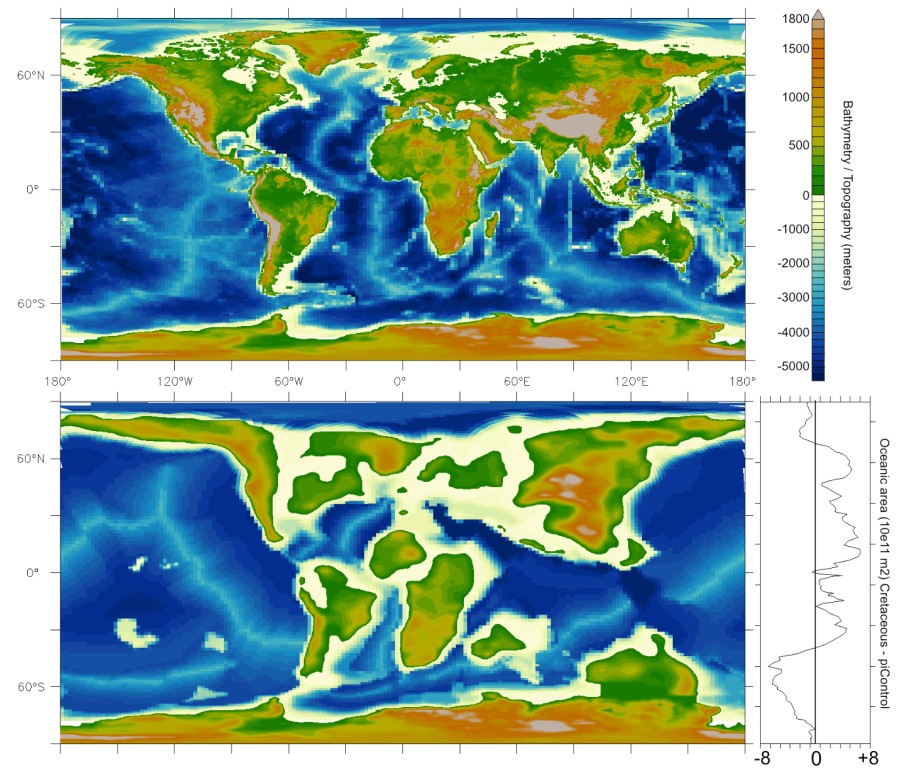

*Figure 3: Evolution of Albedo (surface and planetary) and emissivity, in percentages and of T2M (°C) from piControl to 4X-CRETACEOUS simulations. The major change is always recorded with the change of pCO₂ between 1X-NOICE and 4X-NOICE simulations.*

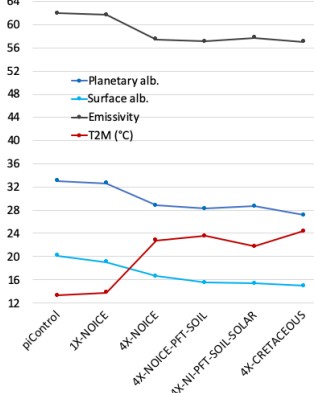



*Figure 4: T2M (°C) for (a) piControl initial simulation and (g) Cretaceous final simulation, and anomalies (°C) for intermediate simulations: (b) 1X-NOICE-piControl, (c) 4X-NOICE-1X-NOICE, (d) 4X-NOICE-PFT-SOIL – 4X-NOICE, (e) 4X-NOICE-PFT-SOIL-SOLAR – 4X-NOICE-PFT-SOIL, (f) 4X-CRETACEOUS - 4X-NOICE-PFT-SOIL.*

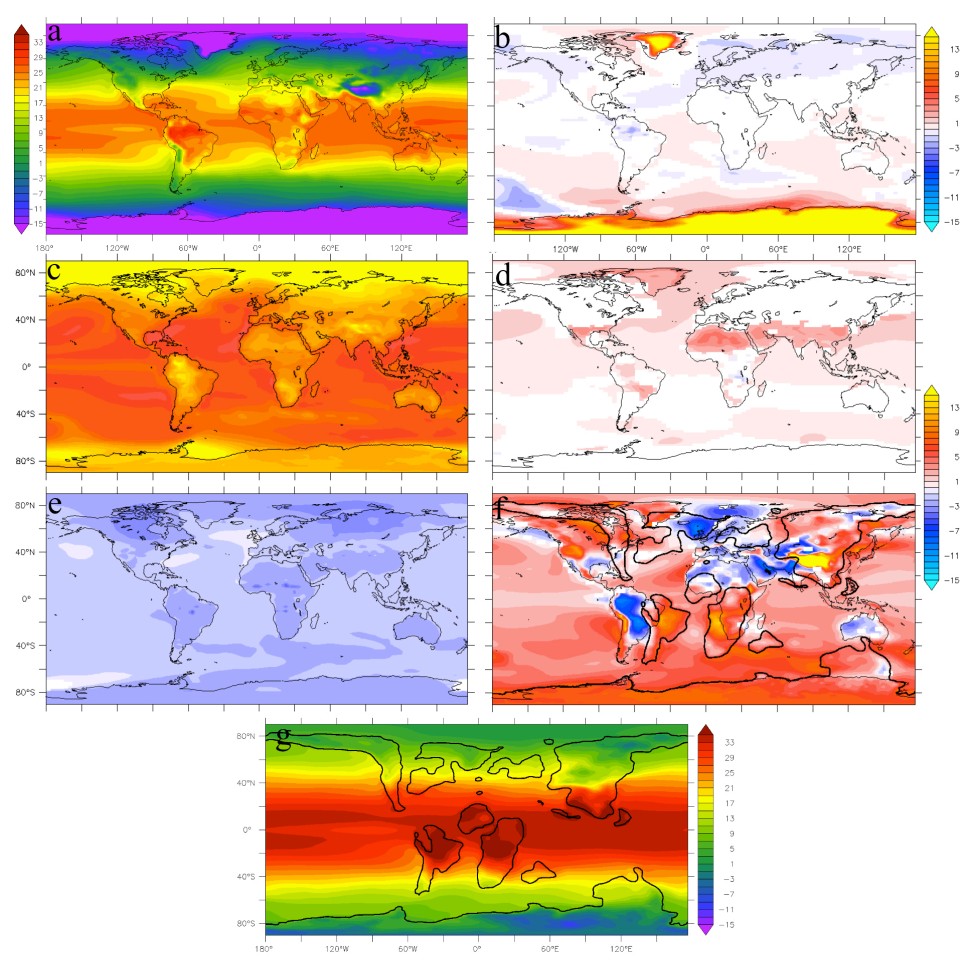


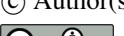




*Figure 5: Cloudiness for 1X-NOICE and 4X-NOICE simulations. (a) Anomaly of total cloudiness (4X-NOICE – 1X NOICE). (b) Low cloudiness (solid curves) and high cloudiness (dashed curves) for 1X-NOICE (black) and 4X-NOICE (red) simulations.*

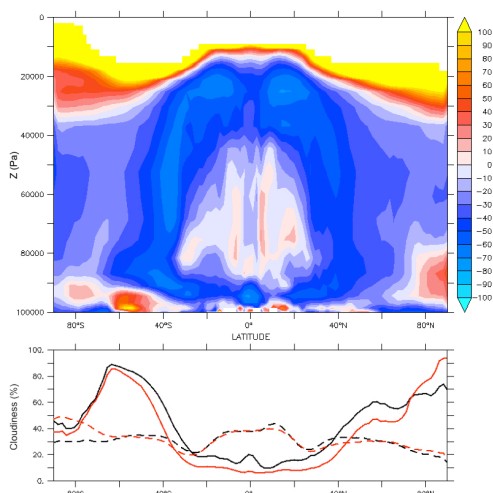

*Figure 6: T2M (°C) meridional gradients for 4X-NI-PFT-SOIL-SOLAR-SOLAR (black) and 4X-CRETACEOUS (red) simulations. Solid curve corresponds to annual average, dashed curves correspond to winter and summer values. The 4X-CRETACEOUS simulation is generally warmer than the 4X-NI-PFT-SOIL-SOLAR-SOLAR simulation, with the exception of the boreal summer.*

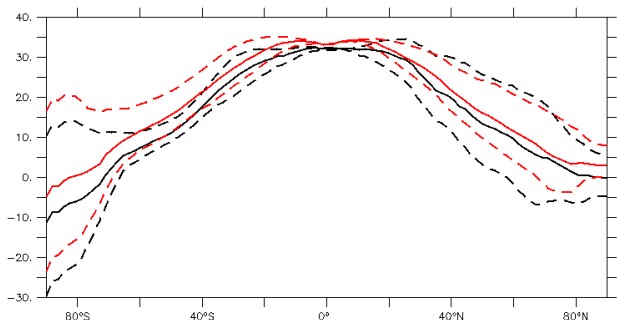




*Figure 7: Surface currents for 4X-NOICE-PFT-SOIL-SOLAR (left) and 4X-CRETACEOUS (right) simulations. (a), (b) Intensity of surface circulation (Sv – Annual Mean for 0-80 meters of water depth). Strong equatorial winds leads to the formation of an equatorial circumglobal current. (c), (d) Intensity of surface circulation (Sv – Annual Mean for 0-80 meters of water depth). The closure of the Drake passage (DP-300 meters of water depth) leads to the suppression of the ACC.*

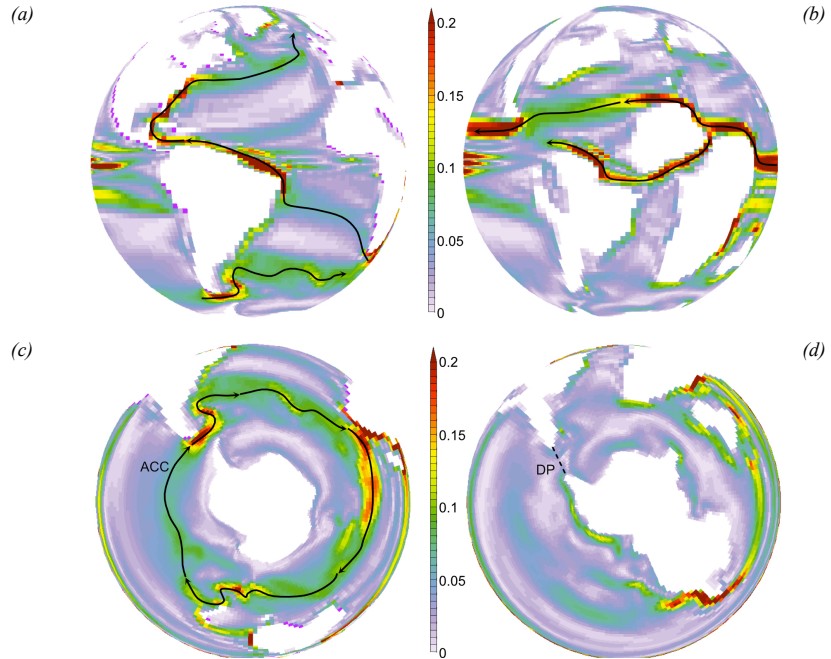













*Figure 8 - (a), (b) Global meridional stream-function (sv) for the first 300 meters of water depth. Red and blue colors indicate*
*clockwise and anti-clockwise circulation respectively. (a): 4X-NI-PFT-SOIL-SOLAR and (b) 4X-CRETACEOUS. (c) Oceanic*
*heat transport for 4X-NI-PFT-SOIL-SOLAR and 4X-CRETACEOUS simulations. Positive and negative values indicate*
*northward and southward transport direction, respectively.*
573       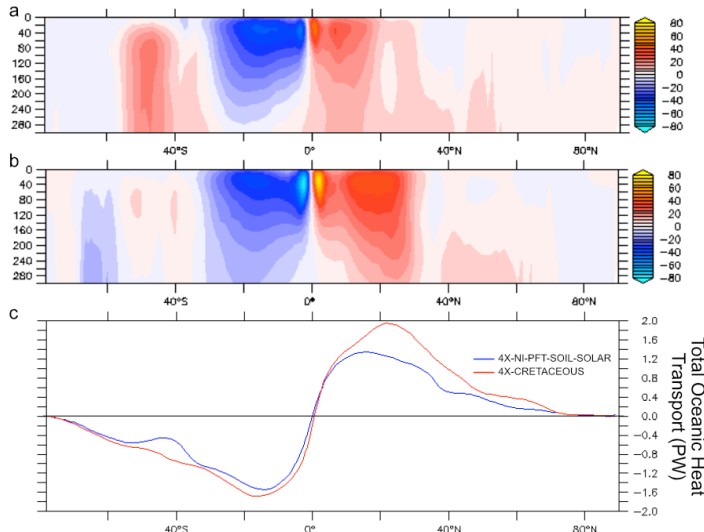
*Figure 9: (a) Meridional Sea-Surface Temperature gradients for all simulations. (b) Same SST curves than (a) but*
*superimposed such as equator temperatures are equal, allowing to compare the steepness of the curves. (c) Meridional*
*atmospheric surface temperature gradients for all simulations. (d) Same curves than (c) but superimposed such as equator*
*temperatures are equal.*

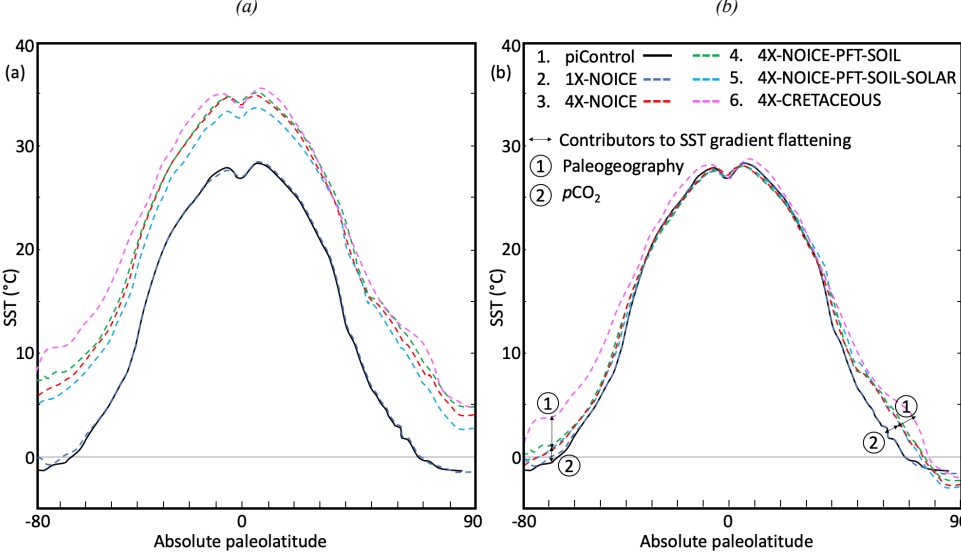



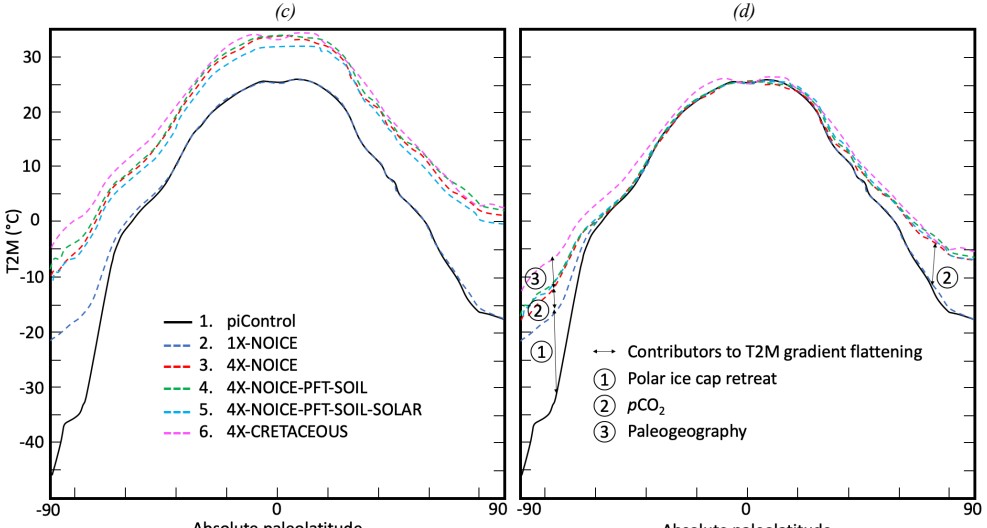

Figure 10: *Meridional surface temperature gradients for the 4X-CRETACEOUS simulation. (a) Oceanic temperatures: the solid line corresponds to the mean annual temperature obtained from the modeling. Dashed lines correspond to winter and summer seasonal averages. The grey shaded areas correspond to local monthly temperatures. The red dashed line correspond to the regression line calculated from data points. Data points are obtained with several proxies for the cenomano-turonian period. The green data point is obtained from TEX 86 for the Maastrichtian (70 Ma) and extrapolated for 90 Ma. The Huber et al. (2018) point is obtained from $\delta^{18}O$ on foraminifera and the Vandenmark et al., 2007 point is interpreted from the presence of crocodilian fossils. MAT=Mean Annual Temperature, CM=Coldest Month. (b) Atmospheric temperatures: same legend as (a) for modelized temperatures. Data points are obtained from several proxies including CLAMP analysis on paleofloras, leaf analyses, paleosol-derived climofunction or bioclimatic analysis. Symbols represent mean annual temperatures and solid lines associated ranges/errors. Dashed lines represent monthly mean temperatures. Orange data points are for cenomano-turonian ages (100-90 Ma), blue data points for Albian and green data points for Coniacian-Santonian (88-85 Ma).*





*Figure 11: Plot of atmospheric and sea surface temperature gradients vs pCO₂ for different modelling studies and data compilation. Data gradients are plotted for a default pCO₂ value of 4 P.A.L. Gradients are expressed in °C per °latitude and are calculated from 30 to 80 degrees of latitude.*

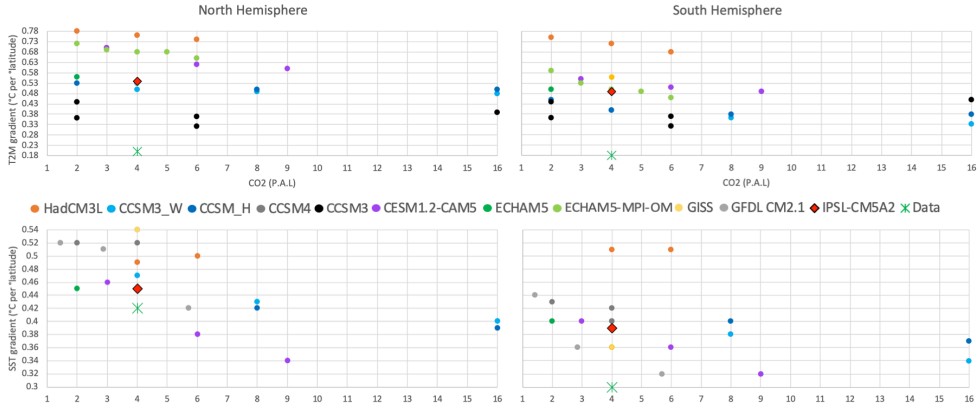

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
