# Peer review of "Stripping back the Modern to reveal the Cenomanian-Turonian climate and"

_Climate of the Past, 2019_

## Referee Comment (RC1) · Anonymous Referee #1 · 25 Feb 2020

**1   General comments**

The authors present a set of six IPSL-CM5A2 Earth System Model simulations, in which the boundary conditions are progressively changed from pre-industrial values to estimates for the Cenomanian-Turonian (94 million years ago). Using these simulations, the authors aim to better understand which of the boundary condition changes caused the reconstructed warm and equable (reduced temperature gradient between low and high latitudes) Cenomanian-Turonian climate.

They find that most of the simulated *global* warming is due to the prescribed 4-fold (relative to pre-industrial) increase in atmospheric $pCO_2$, and that the *temperature gradient* is reduced mostly due to 1) the removal of the Antarctic ice sheet, 2) polar amplification in response to higher $pCO_2$, and 3) enhanced ocean heat transport in response to the different continental configuration.

Unfortunately, the authors do not quite succeed to simulate the reconstructed equable climate, suggesting that the presented list of boundary condition changes and processes that contribute to the more equable climate is not complete or that their contributions are not represented correctly in IPSL.

Although the authors thus do not solve the long-standing problem of simulating equable past climates, the presented results are still interesting and relevant to the field, they are within the scope of CP, and should be published.

However, I do have some corrections and suggestions to improve the manuscript prior to publication, as detailed below.

First, it is a bit unclear to me whether the presented results are supposed to be representative for the Cretaceous in general (as suggested by the title of the manuscript), or for the Cenomanian-Turonian, or for the Oceanic Anoxic Event 2. In the discussion section, due to a lack of other Cenomanian-Turonian simulations, the results are also compared to results for the Eocene, which occurred much later in time than the Cretaceous; but the boundary condition differences between these different time slice simulations are not discussed appropriately.

While the applied approach of successively changing the boundary conditions is appreciated and useful here, and a lot of extra work, the authors do not appropriately discuss any potential shortcomings of the applied linear factorization approach. For example, in a warmer climate like that with $4xCO_2$, a reduction of the surface albedo due to the removal of continental ice sheets would probably have a larger effect, because the surface albedo effect would be masked to a lesser extent by snow cover. I.e., the contributions can depend on the sequence of the changes.

As the conclusions (Section 5) are written now, they are a summary (repetition) of the previously presented results (except for the last two sentences, which to me sound like general speculative remarks). I think the conclusions could be worked out more clearly and in more detail, in particular with the questions in mind that are raised in the introduction. For example, do the results contribute to a consensus on the relative importance of $CO_2$ versus paleogeography (introduction, paragraph starting on page 3 line 78)?

**2 Minor / specific comments**

*Abstract, page 1 line 14:* Please replace "suffered" by something more objective, such as "experienced".

*Abstract, line 17:* As far as I understand the Cretaceous is a period and not an era.

*Page 2, line 55:* What is the "conundrum of Cretaceous $pCO_2$ question"?

*Page 4, line 10:* Are the carbon and biogeochemical cycle models / is PISCES also running in the presented simulations? Will the results be described elsewhere? In this context, any missing feedbacks could be discussed. E.g., in reality, a 4-fold $pCO_2$ increase would certainly also affect the ice sheets.

*Page 5-6, experimental design:* The prescribed boundary condition changes could be described in more detail. How exactly is the ice removed? What happens to the water? Is the ocean salinity adjusted in any of the simulations? What are the properties of the bare soil (e.g., albedo)? What happens to the river routing? To understand the simulated surface warming, an estimate of the pure surface height / lapse rate effect would also be helpful. What are the initial conditions for the $4xCO_2$ runs, why are they only "similar" to those described by Lunt et al.? It is not clear to me how ORCHIDEE works in the presented simulations; are the PFT distributions prescribed and do they

stay the same thereafter in all simulations?

*Page 6, line 61:* TSI in SOLAR-experiment 1351 or 1353W/m$^2$ as in Table 1?

*Page 8, line 189-190:* ...progressive change... induces a general global warming...: this is not true, solar forcing causes cooling.

*Page 8, line 191 and Table 2:* It does not make any sense to me to describe the warming in terms of the percentage of the initial temperature. The total warming should be set to 100% here. The following contributions should be described accordingly, too. That means, for example, $CO_2$ contributes $9°C$ or about 80% to this total warming (and not 61%).

*Page 8-9, Table 2:* The global anomaly rows are somewhat confusing with the different units ($°C$ versus %). Respective rows showing the relative contributions of the changes ($\Delta$ice, $\Delta CO_2$, ...) to the temperature, albedo, emissivity would be useful. That would also help to double-check if the contributions add up to 100%.

*Page 9, lines 214-215:* A decrease of the snow cover over continents probably does not explain the warming over the polar oceans.

*Page 9, line 218-220:* "The relative humidity decrease can be driven by the temperature rise...". Maybe this is just my lack of meteorological knowledge; references?

*Page 9, lines 220-226:* Are the prescribed processes speculative, or have they been identified in the presented simulation?

*Page 10, line 241:* "These contrasted climatic responses to the impact of ice sheets on sea surface temperature have been observed in previous modelling studies but their origin is still unclear..." A discussion of the different suggested mechanisms compared to the processes at work here would be interesting. Is that possible?

*Page 10, lines 259-260:* "These trends are linked to a general decrease of planetary albedo and/or emissivity, ..." Maybe maps of the prescribed surface albedo, actual
surface albedo (with snow and ice), and planetary albedo for some of the different simulations would be helpful to follow the argumentation in this section.

*Page 10, line 261:* Which trend is compensated exactly?

*Page 11, line 267:* The Drake Passage still looks open to me in Fig. 2. Maybe it is better to plot the geographies on the model grid, also showing the sea-land mask?

*Page 12, line 299:* "...increase is due to...", it seems more appropriate to write that it is "consistent with" those previous findings by Rose and Ferreira; or is this evident from the presented simulations?

*Page 12, lines 310-314:* The warming due to the different boundary condition changes does not add up to 100% (49% + 30% - 16% + 4% = 67%). Please check the numbers.

*Page 12, lines 314-317:* "... the increased contribution of paleogeography in the simulated sea surface warming compared to the atmospheric warming, which is probably driven by the major changes simulated in the surface circulation (Fig. 7)." I think this is an interesting hypothesis that could be elaborated in more detail, because the results to test this hypothesis are available here. Regarding the present-day North Atlantic, I suspect that the large cooling patch in that area in the CT-simiulation (Fig. 4f) could be due to the lack of a Gulf Stream or North Atlantic current equivalent. I.e., in that case, the changed circulation leads to a high-latitude cooling rather than to a warming.

*Page 12, line 320:* How exactly are the meridional temperature gradients calculated? $(T(30°)-T(80°))/50°$, without any averaging applied? Why between $30°$ and $80°$, and not between equator and poles?

*Page 12, line 325:* ...the SST gradient can be visualized (not explained) by...

*Page 13, line 355:* To me it looks like the local warming due to the removal of the Greenland ice sheet would have a visible effect on the meridional temperature gradient, but this effect is somewhat counterbalanced by the cooling in Europe and Siberia. As mentioned earlier, I think it would be interesting to elaborate on the processes at work

here (e.g., stationary wave feedback?).

*Page 13, lines 360-362:* I do not agree. Looking at Fig. 4f, the warming at 40-70°S appears to be more pronounced in the proto-Indian Ocean than over proto-Australia, i.e., the enhanced warming is probably not due to the larger continental area.

*Page 13, lines 362-363:* Again, I don't agree that this should be only an effect of the ocean area (but also, for example, due to the lack of a Gulf Stream or North Atlantic current equivalent).

*Page 14, Section 4.1* A large part of this so-called discussion section is a comparison of the presented model results to previously published proxy data – the section should probably be titled accordingly.

*Page 14, line 369:* This is not really a prediction, but rather a hindcast; "were compared" meaning "are here compared"?

*Page 14, line 370-371:* What exactly does "essentially based on Tabor et al. 2016" mean? What are the differences?

*Page 14, line 383:* Comparison shown in Fig. 10, not 9a?

*Page 14, lines 393-394:* It could make the discussion of the data-model fit and of the wide spread of temperatures in one latitudinal band easier if the proxy data was also shown on a 2D map (like Fig. 4f). The DSDP locations could then also be marked.

*Page 14, line 400:* Between 25 and about 28 (not 30)°C according to Fig. 10a?

*Page 15, lines 407-409:* This is a main point and conclusion of the study, and should be worked out in more detail. "...we observe the same underestimate...", same as whose / same as what? The temperatures are compared to proxy data, not observations.

*Page 15, lines 410-416:* How exactly are the gradients computed from proxy data? Please add a measure of uncertainty, the range of the reconstructed temperatures for most latitudinal bands is very large.

*Page 15, lines 424-425:* "... with the lowest latitudinal gradients being obtained for the highest $pCO_2$ values." This is not correct according to Fig. 11. The lowest gradient seems to be computed for the $6xCO_2$ CCSM3 simulation. Also, for example, the $2xCO_2$ Eocene simulation with ECHAM5/MPIOM shows a lower northern hemisphere temperature gradient than any of the 2, 3, 4, 5, or 6x Cretaceous simulations with ECHAM5/MPIOM. Note that the simulation denoted here as ECHAM5 is actually also an ECHAM5/MPIOM run. This illustrates that the differences in the applied boundary conditions between the simulations other than $CO_2$ play a large role and should be discussed in detail.

*Page 16, line 441:* The title of the subsection is a bit misleading, since not only Cretaceous studies are discussed (also e.g. Eocene).

*Page 16, lines 446-448:* 2000 years are not very long for equilibrium climate sensitivity simulations, and the $4xCO_2$ simulations do still show a significant trend at the end (SST cooling by about 1K/thousand years; Fig. 1); this should be pointed out here.

*Page 16, line 469:* Please rephrase, "reduced gradient was amplified"?

*Page 17, line 473:* Please rephrase, "is the primary control" -> "is a primary control"; other similarly large effects may be missing in the model (given the model–proxy data mismatch).

*Page 17, lines 484-488:* What would be the probable effects of such a vegetation change, and what the implications for this study?

*Page 18, lines 512-513:* "Such modelling efforts would probably even more increase the equilibrium climate sensitivity, ..." This is interesting but speculative. Please delete or discuss in more detail / add references.

*Page 18, data availability:* Are the model source code and the applied boundary condition files also available?

*Figure 1:* Labels are too small.

*Figure 2:* Ticks and labels of oceanic area anomaly plot do not line up.

*Figure 3:* Labels are too small. Fewer contour intervals might be better, also to identify the zero-line. Maybe this is just my printer, but I don't see white in the colorbar.

*Figure 5:* Labels are way too small, especially in the lower panel. I do not know by hard what the usual cloud cover is at what pressure level. Plots of the total cloudiness from observations and from PI and maybe also 1x-NOICE would be helpful. Is the anomaly really plotted as the absolute difference in cloud cover, as suggested in the caption? That would indicate that the top of the simulated atmosphere is totally covered by clouds!?! Please also define in the text what is meant by high and low clouds.

*Figure 6:* Caption: "4xNI ... SOLAR-SOLAR" typo experiment name.

*Figure 9b+d:* Instead of shifting the curves to the same equatorial temperatures, it might be better to plot the anomalies due to the individual boundary condition changes ($\Delta CO_2$, ...).

*Figure 10:* The red dashed regression line is not visible in my copy.

*Figure 11:* Labels are too small.

**3  Language / typos**

There are numerous typos and little grammatical errors, as well as some unclear sentences; I would hence recommend copy editing.

For example: *Page 3, line 85:* ...latest "work" are divided...  *"studies" instead?*

*Page 4, line 96:* ...a set of simulation run... (missing plural s)

*Page 9, lines 208-209:* The whole surface is warmer... and which is generally larger over continents... (sentence structure)

*Page 10, line 251:* ...data ant temperature... (and)

---

## Referee Comment (RC2) · Alexander Farnsworth (Referee) · 2 Mar 2020

Overview:

This paper attempts to deconvolve the impact of four different boundary conditions (carbon dioxide, ice sheets, solar luminosity and vegetation/soil parameters) on the climate and temperature gradients of the Late Cretaceous to understand their relative roles. Through the use of the IPSL model a series of simulations are performed with changing the model boundary conditions in a step-wise fashion through introducing each one until a fully Late Cretaceous is simulated from the Pre-industrial.

[Figure]

They find that CO2 has the largest impact on the climate signal, but also signify that paleogeography also plays an important role in perturbations in the pole-equator temperature gradient though changes in ocean circulation and feedbacks.

General comments:

This is an interesting study using an updated version of the IPSL model looking at the long standing issue of equable climates of the past problem. I like the experiment design used within this study, it nicely deconvolves the different boundary conditions being investigated in a somewhat clean manner. Although it is, as I am sure the authors well know, more difficult to identify the direct and indirect influence of the non-linear feedbacks that can also occur without a full suite of simulations looking at each boundary condition individually/in pairs in a matrix style approach.

Although the issue is still present this work does provide some interesting results and what influence each boundary condition investigated has on the general climate and the pole-equator temperature gradient.

I think this study should be published and I look forward to being able to reference it, however I do have some comments that may help improve the manuscript before publication.

• Why should the Cenomanian-Turonian thermal maximum be of more relevance than that of the PETM, MECO, etc. to the future. Worth while to flesh out why this particular time period is deemed important • There will also be non-linear feedbacks that cannot be deconvolved with the current methodology. I.e. The proportion of landmass at different latitudes between the pre-industrial and Cretaceous at Preindustrial CO2 levels (and topographic height differences) will have a dissimilar impact on the vegetation response and in turn the warming response from increased CO2. • I did sometimes get a bit confused with the methodology section. It was sometimes hard to see what models were initialised with what initial conditions. How were some of the boundary conditions treated. i.e. what were the actual albedo prescribed? Was the

vegetation uniformly applied even to mountain regions where it may not grow? Might be worth being a bit more concise in describing them individual boundary conditions and how they were implemented for clarity and reproducibility. • Minor point. Some of the use of English could be improved in places, however this is fairly ancillary and does not in any way detract from the good science being shown. • Page 5 – Line 40. It looks like only the 1X-NOICE is in equilibrium in the deep ocean. The other four still appear to be trending. I think for clarity this should be stated or changed to "near-surface equilibrium". I certainly sympathise as some of my own simulations can take up to 10,000 model years to reach equilibrium. However, I do not think this will change the overall results, but for clarity it should not be stated as complete equilibrium. Gregory plots may also be another useful diagnostic to see if you have an energy imbalance otherwise. • Ice sheet removal impact. I agree with your assessment of the regional impact, however it might also be that you get a pseudo ice sheet in the 1X-NOICE simulation with perennial snow cover over the soil surface, just a low elevation one. I suspect this is the case as in the 4X-NOICE you get a much large response in the change in surface and planetary albedo. Did you ever run a 4xCO2 experiment with ice sheets in the pre-industrial to see the relative impact of just the CO2? • Do you see any change in ocean circulation patterns from removal of the ice sheet and increased CO2 in the preindustrial?

Minor comments:

• Page 1 & 2 – Line 17 & 33. 'period' not "era" • I think it is worthwhile to define what you mean by 'high' and 'low' latitudes as you often see different values purported in different studies for clarity. • Throughout – put references in chronological order. • Page 2, Line 60. Define "P.A.L." here. This is the first instance in the ms rather than on page 6- line 50. • Page 3 – Line 79. Change sentence to "the primary driver of Cretaceous climate has been suggested". • Page 3 – Line 83. Delete erroneous "s". • Page 4 – Line 96. Probably more accurate to say "We performed six simulations using both Pre-industrial and Late Cretaceous boundary conditions where we

incrementally modify the Pre-industrial boundary conditions to that of the Late Creta-ceous for…..1,2,3,4". • Page 4. Although stated that IPSL-CM5A2 has been used for contemporary and future climate simulations it would be worth adding a line that states how well the model performs in simulating a modern-day climate. • Page 5 – Line 20. Repetition of "long" in the sentence. Remove one of them. • Page 5 – Line 37. "retreat" to retreat. But I think 'removed' would be more accurate. • Page 6 – Line 53. Is that from the 4X-NOICE simulation? • Page 6 – Line 218-222. Did you look at atmospheric stability arguments in relation to this? • Page 6 – Line 223. Do you mean greater season ice melt or there being less sea ice area in the 4X-NOICE compared to the 1X-NOICE simulation? • Page 12 – Section 3.5.1. The percentage change adds to 99%. Rounding error? • Page 12 –Line 321-324: There does not appear to be any change in the N.Hem SST gradients (0.45/lat). Any idea why? You attribute the Greenland ice sheet/sea ice for less sensitivity in the atmospheric gradient of the N.Hem. Something similar here? • Page 14 – Line 370. Do you mean that you used the Tabor, et al. dataset and adding more data points to it? • Page 14 – Line 388. Only if you suggest there is a seasonal proxy bias. This is mentioned later on, but might be worth a few ref's that show there are seasonal bias in some proxies. • Page 16 – Line 448. Do you mean 'more complex' rather than "large"? • Page 17 – Line 480. "cloud" not "clouds". • Page 17 – Conclusions section. This is a tad bit repetitive of the results/discussion. Perhaps broaden this out with respect to the dis-cussion in your introduction. • Figure 1, 5,6, 8, 9, 11 Captions. Add 'mean annual' to caption. • Figure 2. Is that the model resolution geographies? • Figure 11b. Appears to be some modelling studies missing? E.g. HadCM3L 2xCO2 data point and others. Or was I interpreting this wrongly? Quite possible!

Best, Alex Farnsworth
* * *

---

## Author Comment (AC1) · 8 Apr 2020

**1 General comments**

First, it is a bit unclear to me whether the presented results are supposed to be representative for the Cretaceous in general (as suggested by the title of the manuscript), or for the Cenomanian-Turonian, or for the Oceanic Anoxic Event 2.

→ The results presented here are representative only for the Cenomanian-Turonian. Simulations for Early Cretaceous or Late Cretaceous (Maastrichtian) would imply different boundary conditions (e.g. paleogeography). It may be not clear enough in the

manuscript, we will clarify it by changing "Cretaceous" to "Cenomanian-Turonian" in the title and in the manuscript.

In the discussion section, due to a lack of other Cenomanian-Turonian simulations, the results are also compared to results for the Eocene, which occurred much later in time than the Cretaceous; but the boundary condition differences between these different time slice simulations are not discussed appropriately.

→ Indeed, the boundary conditions are different for the Eocene and the comparison of gradients cannot be done directly. The idea was more to have a discussion on the temperature gradients simulated by Earth System Models in general.

While the applied approach of successively changing the boundary conditions is appreciated and useful here, and a lot of extra work, the authors do not appropriately discuss any potential shortcomings of the applied linear factorization approach. For example, in a warmer climate like that with 4xCO2, a reduction of the surface albedo due to the removal of continental ice sheets would probably have a larger effect, because the surface albedo effect would be masked to a lesser extent by snow cover. I.e., the contributions can depend on the sequence of the changes.

→ You are correct. We agree that repeating the experiments with a different sequence of changes would be desirable to fully assess the shortcomings of the linear factorization method. It indeed has been suggested that a different sequence of boundary condition changes would probably give different results (Lunt et al., 2012). However, these additional experiments using the IPSL-CM5A2 earth system model require a computational cost that we cannot afford. We have added the following discussion about linear factorization in the revised manuscript: "The choice of applying a linear factorization approach was made for problems of computing time and cost. Changing the sequence of changes or applying a symmetrical factorization as in Lunt et al (2012) would require too many supplementary simulations. However, such a method is very dependent on the sequence of changes, and results would be probably different if

boundary conditions were change in a different order (Lunt et al., 2012)".

As the conclusions (Section 5) are written now, they are a summary (repetition) of the previously presented results (except for the last two sentences, which to me sound like general speculative remarks). I think the conclusions could be worked out more clearly and in more detail, in particular with the questions in mind that are raised in the introduction. For example, do the results contribute to a consensus on the relative importance of CO2 versus paleogeography (introduction, paragraph starting on page 3 line 78)?

→ We will rearrange the conclusion in order to be less repetitive with the previous sections, but also to be more coherent with the questions raised in the introduction.

2 Minor / specific comments Abstract, page 1 line 14: Please replace "suffered" by something more objective, such as "experienced".

→ Done.

Abstract, line 17: As far as I understand the Cretaceous is a period and not an era.

→ We replaced Âń era Âż by Âń period Âż where needed throughout the revised manuscript.

Page 2, line 55: What is the "conundrum of Cretaceous pCO2 question"?

→ We meant that atmospheric pCO2 levels are not well constrained in the Cretaceous with a large spread in the values inferred from proxy data (e.g. Wang et al. ESR 2014, Foster et al. Nat Comm. 2017). The sentence was rephrased as follows: "Modelling studies have also focused on estimating Cretaceous atmospheric CO2 levels (Barron et al., 1995; Poulsen et al., 2001, 2007; Berner, 2006; Bice et al., 2006; Monteiro et al., 2012) in an attempt to refine the large spread in values inferred from proxy data (<900 to >5000 ppm)."

Page 4, line 10: Are the carbon and biogeochemical cycle models / is PISCES also

running in the presented simulations? Will the results be described elsewhere? In this context, any missing feedbacks could be discussed. E.g., in reality, a 4-fold pCO2 increase would certainly also affect the ice sheets.

→ PISCES is also running but results will be presented in another paper in the near future. Based on previous published results by members of our group (Ladant and Donnadieu, 2016), we have considered that no ice-sheet could be formed at 4-fold pCO2 during the C-T.

Page 5-6, experimental design: The prescribed boundary condition changes could be described in more detail. How exactly is the ice removed? What happens to the water? Is the ocean salinity adjusted in any of the simulations? What are the properties of the bare soil (e.g., albedo)? What happens to the river routing? To understand the simulated surface warming, an estimate of the pure surface height / lapse rate effect would also be helpful. What are the initial conditions for the 4xCO2 runs, why are they only "similar" to those described by Lunt et al.? It is not clear to me how ORCHIDEE works in the presented simulations; are the PFT distributions prescribed and do they stay the same thereafter in all simulations?  → We will rearrange the description of boundary conditions to improve clarity and readability. In simulations with a preindustrial geography, polar ice sheets were removed by setting the fixed land ice fraction to 0 in the model. The topography has been changed accounting for isostatic rebound resulting from the loss of the land ice beneath the Greenland and Antarctic ice sheets. In place of the ice, soil and vegetation parameters were prescribed to brown bare soil (no vegetation), for which the albedo is derived from the soil color (Krinner et al., 2005). River routing stays unchanged. We will add supplementary figures of the new topography without ice sheets and of the temperature change due to the lapse rate effect. Oceanic initial conditions in simulations with 4x CO2 concentrations are adapted from Lunt et al., 2017. In fact, the constant initial salinity field (to 34.7 PSU) is identical to that proposed by Lunt et al. 2017 but the initial ocean temperature field is slightly different from that of Lunt et al. 2017 because of stability issues in the model.

We therefore use the following temperature conditions (see also Sepulchre et al., GMD 2020, in review) :

If depth ≤ 1000 m: T=10+((1000-depth)/1000)*25 cos⁡(latitude)

if depth > 1000 m : T = 10 In our simulations, vegetation in ORCHIDEE is semi-dynamic in the sense that PFTs are prescribed and cannot change but plant phenology is predicted by the model based on surface climate conditions.

Page 6, line 61: TSI in SOLAR-experiment 1351 or 1353W/m2 as in Table 1?

→ Corrected to 1353 W/m2.

Page 8, line 189-190: ...progressive change... induces a general global warming...: this is not true, solar forcing causes cooling.

→ Indeed, the change in solar forcing induces a cooling as indicated on line 195. The general global warming mentioned here refers to the global temperature change in the last simulation (4X-Cretaceous) relative to the first simulation (preindustrial), that is, the summed effect of all boundary condition changes.

Page 8, line 191 and Table 2: It does not make any sense to me to describe the warming in terms of the percentage of the initial temperature. The total warming should be set to 100% here. The following contributions should be described accordingly, too. That means, for example, CO2 contributes 9°C or about 80% to this total warming (and not 61%).

We agree that this section was confusing and we will change it in the revised manuscript. To include the impact of solar forcing (cooling), which is opposite to those of other boundary condition changes, we had chosen to describe the temperature changes in terms of percentage of the cumulative absolute temperature change. However, as it is confusing, we decided to remove the percentages and to keep only the raw temperature values.

[Figure]

Page 8-9, Table 2: The global anomaly rows are somewhat confusing with the different units (_C versus %). Respective rows showing the relative contributions of the changes (ice, CO2, ...) to the temperature, albedo, emissivity would be useful. That would also help to double-check if the contributions add up to 100%.

→ We removed the percentages that were expressed in terms of initial value percentage and keep only the raw values to avoid confusions.

Page 9, lines 214-215: A decrease of the snow cover over continents probably does not explain the warming over the polar oceans.

→ Modified sentence: "First, a decrease of sea ice and snow cover (especially over Northern hemisphere continents and along the coasts of Antarctica) leading to surface albedo decrease, explains the warming amplification over polar oceans and continents."

Page 9, line 218-220: "The relative humidity decrease can be driven by the temperature rise...". Maybe this is just my lack of meteorological knowledge; references?

→ This part of the manuscript is indeed not very clear. We would need to go into details regarding the changes in water content, relative/specific/absolute humidity as well as atmospheric dynamics to explain the observed changes. We do not want to go into such details in the manuscript which is already long and which not specifically focus on the impact of a CO2 increase, so we finally decided to remove this paragraph.

Page 9, lines 220-226: Are the prescribed processes speculative, or have they been identified in the presented simulation? → The processes are identified in the simulation, however, as explained for the previous comment, we finally removed this paragraph from the manuscript.

Page 10, line 241: "These contrasted climatic responses to the impact of ice sheets on sea surface temperature have been observed in previous modelling studies but their origin is still unclear..." A discussion of the different suggested mechanisms compared

to the processes at work here would be interesting. Is that possible? → We do not detail here the processes responsible for the contrasted climatic response to the ice sheet absence (or presence) as the simulated impacts are mostly regional, and because the climatic consequences of the absence/presence of ice sheets at the poles are already the focus of other studies (Goldner et al., 2014; Knorr and Lohmann, 2014; Kennedy et al., 2015; Kennedy-asser et al., 2020). In these studies, as well as in our simulations, local changes in winds are observed, probably due to topographic changes linked to the existence of the ice sheet. These changes in wind may locally impact oceanic currents, deep-water formation and oceanic heat transport, and thus explain the regional contrasts in temperature changes around the Antarctic continent. We added some precisions in the manuscript to briefly describe these processes:

"These contrasted climatic responses to the impact of ice sheets on sea surface temperatures have been documented in previous modeling studies (Goldner et al., 2014; Knorr and Lohmann, 2014; Kennedy et al., 2015). Their origin in the Southern Ocean is still unclear but, in these studies as well as in our simulations, it seems that local changes in winds, due to topography change after polar ice cap removal, are impacting locally oceanic currents, deep-water formation and thus oceanic heat transport and temperature distribution. In the Northern Hemisphere, the observed cooling over Eurasia could be linked to stationary wave feedbacks due to the change in topography after Greenland ice cap removal (See supplementary information; see also Maffre et al., 2018)."

Page 10, lines 259-260: "These trends are linked to a general decrease of planetary albedo and/or emissivity, ..." Maybe maps of the prescribed surface albedo, actual surface albedo (with snow and ice), and planetary albedo for some of the different simulations would be helpful to follow the argumentation in this section.

→ Supplementary figures of planetary and surface albedo as well as emissivity will be provided.

Page 10, line 261: Which trend is compensated exactly?

→ The increase in albedo, which drives cooling during summer, is compensated by a strong emissivity decrease during winter, leading to winter warming. Modified sentence in the manuscript: "This increase in albedo is compensated by a strong atmosphere emissivity decrease during winter but not during summer, which leads to the seasonal pattern of cooling and warming (See Supplementary Figure S5)."

Page 11, line 267: The Drake Passage still looks open to me in Fig. 2. Maybe it is better to plot the geographies on the model grid, also showing the sea-land mask?

→ Indeed, the Drake Passage is not completely closed but is very shallow (<300 meters of water depth). This configuration essentially shuts down the ACC, as observed on figure 7.

Page 12, line 299: "...increase is due to...", it seems more appropriate to write that it is "consistent with" those previous findings by Rose and Ferreira; or is this evident from the presented simulations?

→ Modified sentence: "This high-altitude cloudiness increase is consistent with the extratropical increase in OHT…"

Page 12, lines 310-314: The warming due to the different boundary condition changes does not add up to 100% (49% + 30% - 16% + 4% = 67%). Please check the numbers.

→ The 30% is wrong and should be 31%. However, we finally decided to remove the percentages as it was confusing.

Page 12, lines 314-317: "... the increased contribution of paleogeography in the simulated sea surface warming compared to the atmospheric warming, which is probably driven by the major changes simulated in the surface circulation (Fig. 7)." I think this is an interesting hypothesis that could be elaborated in more detail, because the results to test this hypothesis are available here. Regarding the present-day North Atlantic, I suspect that the large cooling patch in that area in the CT-simulation (Fig. 4f) could be

Interactive
comment

due to the lack of a Gulf Stream or North Atlantic current equivalent. I.e., in that case, the changed circulation leads to a high-latitude cooling rather than to a warming.

→ The cooling patch observed in the Northern part is located over the Greenland and is due to ocean becoming land in the CT simulation. The cooling patch observed in the Arctic (North of Greenland) could be indeed due to changes in circulation in the North Atlantic. However, this is a regional effect that does not dominate the global signal. The Cretaceous simulation is globally warmer at these latitudes regarding the latitudinal temperature gradients for both winter and summer seasons (fig 6). We do not want to go into details for such regional effects, as they are too numerous to explain, and we want to stay focus on the global signals.

Page 12, line 320: How exactly are the meridional temperature gradients calculated? (T(30_)-T(80_))/50_, without any averaging applied? Why between 30_ and 80_, and not between equator and poles?

→ The meridional gradients are indeed calculated by doing (T(30)-T(80))/50. No average is applied, the values $30°$ and $80°$ where chosen to be coherent with the Upchurch et al., 2015 study.

Page 12, line 325: ...the SST gradient can be visualized (not explained) by...

→ Modified sentence: "The progressive flattening of the SST gradient can be visualized by superimposing the zonal mean temperatures of the different simulation and by adjusting them at the Equator (Fig 9b)."

Page 13, line 355: To me it looks like the local warming due to the removal of the Greenland ice sheet would have a visible effect on the meridional temperature gradient, but this effect is somewhat counterbalanced by the cooling in Europe and Siberia. As mentioned earlier, I think it would be interesting to elaborate on the processes at work here (e.g., stationary wave feedback?).

→ Indeed, the observed cooling can be due to modifications of stationary waves, at

least for the continental part of Eurasia (see figure below). This cooling is however very weak (<1.5°C) compared to the warming observed over Greenland (up to 20°C). We added the precision to the manuscript and will add the following figures to supplementary information:

"In the Northern Hemisphere, the observed cooling over Eurasia could be linked to stationary wave feedbacks due to the change in topography after Greenland ice cap removal (See supplementary information; see also Maffre et al., 2018)."

See Figure 1 in the author response.

Page 13, lines 360-362: I do not agree. Looking at Fig. 4f, the warming at 40-70_S appears to be more pronounced in the proto-Indian Ocean than over proto-Australia, i.e., the enhanced warming is probably not due to the larger continental area.

→ Indeed, the warming is more pronounced over ocean than over proto-Australia. This warming can be explained by the increased southward oceanic heat transport (cf Results - p.11 – line 277). We modified the paragraph to include this result: "In the mid- to high-latitudes of the Southern Hemisphere, the increase in poleward OHT and in the continental area fraction in the CT drive a reduction in the steepness of the atmospheric gradient."

Page 13, lines 362-363: Again, I don't agree that this should be only an effect of the ocean area (but also, for example, due to the lack of a Gulf Stream or North Atlantic current equivalent).

→ Here, we analyze the global signal, and do not look individually at the different basins. Considering that the global ocean heat transport stays stronger for the Cretaceous (even without Gulf Stream - figure 8c), we don't rely changes in circulation to a lack of paleogeography impact on atmospheric gradient flattening.

Page 14, Section 4.1 A large part of this so-called discussion section is a comparison of the presented model results to previously published proxy data – the section should

probably be titled accordingly.

→ New title : 4.1 ABOUT THE CENOMANIAN-TURONIAN CLIMATE : MODEL/DATA COMPARISON

Page 14, line 369: This is not really a prediction, but rather a hindcast; "were compared" meaning "are here compared"?

→ Modified sentence: "The results predicted by our CT simulation are here compared to reconstructions of atmospheric and oceanic paleotemperatures inferred from proxy data (Fig 10a,b)."

Page 14, line 370-371: What exactly does "essentially based on Tabor et al. 2016" mean? What are the differences?

→ Most of the data used here for the comparison come from Tabor 2016, and we added some other points from more recent studies. All the references are indicated in the supplementary data. → Modified sentence: "Our SST data compilation is modified from Tabor et al (2016), with additional data from more recent studies (see Supplementary data)."

Page 14, line 383: Comparison shown in Fig. 10, not 9a?

→ Yes

Page 14, lines 393-394: It could make the discussion of the data-model fit and of the wide spread of temperatures in one latitudinal band easier if the proxy data was also shown on a 2D map (like Fig. 4f). The DSDP locations could then also be marked.

→ We are working here in terms of latitudinal gradients, that's why we didn't want to make the 2D map that would give another direction to the manuscript.

Page 14, line 400: Between 25 and about 28 (not 30)_C according to Fig. 10a?

→ Yes

Page 15, lines 407-409: This is a main point and conclusion of the study, and should be worked out in more detail. "...we observe the same underestimate...", same as whose / same as what? The temperatures are compared to proxy data, not observations.

→ Modified sentences: "The simulated annual SSTs reach a monthly maximum of 28°C around the location of site DSDP 258. We speculate that a seasonal bias in the foraminiferal record may represent a possible cause for this difference, as may local deviations of the regional seawater d180 from the globally assumed -1‰ value. The same trend is observed for atmospheric temperatures with data indicating higher temperatures than the model at high latitudes in both the Southern and Northern Hemispheres. This inter-hemispheric symmetry in model-data discrepancy could indicate a systematic cool bias of the simulated temperatures."

Page 15, lines 410-416: How exactly are the gradients computed from proxy data? Please add a measure of uncertainty, the range of the reconstructed temperatures for most latitudinal bands is very large.

→ The gradients are calculated based on the slope of the linear regression line (see supplementary data 1). We added in the supplementary file the standard error for the regression line.

Page 15, lines 424-425: "... with the lowest latitudinal gradients being obtained for the highest pCO2 values." This is not correct according to Fig. 11. The lowest gradient seems to be computed for the 6xCO2 CCSM3 simulation. Also, for example, the 2xCO2 Eocene simulation with ECHAM5/MPIOM shows a lower northern hemisphere temperature gradient than any of the 2, 3, 4, 5, or 6x Cretaceous simulations with ECHAM5/MPIOM. Note that the simulation denoted here as ECHAM5 is actually also an ECHAM5/MPIOM run. This illustrates that the differences in the applied boundary conditions between the simulations other than CO2 play a large role and should be discussed in detail.

→ Indeed, this observation is only true when considering the same model and boundary conditions (except for the SST of the HadCM3L), but is not correct when considering the different boundary conditions (i.e Eocene vs Cretaceous for the ECHAM5 and CCSM3). We also have different meridional gradients for the same model/boundary conditions but with different model parametrization (e.g. altered cloud physics). We made the correction in the manuscript: "However, when comparing different studies with the same model (Cretaceous vs Eocene for ECHAM5 and CCSM3) it is not the case, indicating the major role of boundary conditions in defining the latitudinal temperature gradient. For instance, the South Hemisphere gradient obtained for the Eocene with the ECHAM5 model is always lower than those obtained for the Cretaceous with the same model, regardless of the $pCO2$ value (See Fig. 11 and Supplementary Data)."

Page 16, line 441: The title of the subsection is a bit misleading, since not only Cretaceous studies are discussed (also e.g. Eocene).

→ Modified title: 4.2 PRIMARY CLIMATE CONTROLS

Page 16, lines 446-448: 2000 years are not very long for equilibrium climate sensitivity simulations, and the 4xCO2 simulations do still show a significant trend at the end (SST cooling by about 1K/thousand years; Fig. 1); this should be pointed out here.

→ We have clarified this point: - Page 5, line 40: "The piControl simulation was run for 1800 years and the five others for 2000 years in order to reach a near-surface equilibrium (Fig.1)". - Page 16 line 448: "The signal is notably due to a 9°C warming in response to the fourfold increase in $pCO2$, which converts to an increase of 4.5°C for a doubling of $pCO2$ (assuming that the response is linear). This sensitivity agrees with the higher end of the range of values in the investigations mentioned above. However, the sensitivity of IPSL-CM5A2 in our simulations could be slightly lower as the simulations are not completely equilibrated – see Figure 1)".

Page 16, line 469: Please rephrase, "reduced gradient was amplified"?

→ Modified sentence : It has been suggested that high latitude warming was amplified in deep time simulations by rising CO2 via cloud and vegetation feedbacks (Otto-bliesner and Upchurch, 1997; Deconto et al., 2000) or by increasing ocean heat transport (Barron et al., 1995; Schmidt and Mysak, 1996; Brady et al., 1998), in particular when changing the paleogeography (Hotinski and Toggweiler, 2003), thus contributing to the flattening of the meridional SST gradient".

Page 17, line 473: Please rephrase, "is the primary control" -> "is a primary control"; other similarly large effects may be missing in the model (given the model–proxy data mismatch).

→ Ok

Page 17, lines 484-488: What would be the probable effects of such a vegetation change, and what the implications for this study?

→ Modifications: "The presence of such a type of vegetation could change the albedo of continental regions, but also heat and water-vapor transfer by modified evapotranspiration processes, thus leading to a warming amplification at high-altitudes and thus a reduced temperature gradient (Otto-bliesner and Upchurch, 1997; Hay et al., 2019). Based on these studies and on our results, we cannot exclude that this kind of high latitude vegetation can give more weight to the role of paleovegetation in reducing the temperature gradient".

Page 18, lines 512-513: "Such modelling efforts would probably even more increase the equilibrium climate sensitivity, ..." This is interesting but speculative. Please delete or discuss in more detail / add references.

→ Deleted.

Page 18, data availability: Are the model source code and the applied boundary condition iles also available?

→ Yes, they are, we added the precisions to the manuscript.

Figure 1: Labels are too small. Figure 2: Ticks and labels of oceanic area anomaly plot do not line up.

→ Modified. Figure 4: Labels are too small. Fewer contour intervals might be better, also to identify the zero-line. Maybe this is just my printer, but I don't see white in the colorbar.

→ Labels and contour intervals modified. We also added in the figure legend that the white color (which is not in the colorbar) corresponds to areas where the anomaly is not significative regarding the student test.

Figure 5: Labels are way too small, especially in the lower panel. I do not know by hard what the usual cloud cover is at what pressure level. Plots of the total cloudiness from observations and from PI and maybe also 1x-NOICE would be helpful. Is the anomaly really plotted as the absolute difference in cloud cover, as suggested in the caption? That would indicate that the top of the simulated atmosphere is totally covered by clouds!?! Please also define in the text what is meant by high and low clouds.

→ We added in the figure legend and in the text, the level pressure for low-altitude cloudiness (>680 hPa) and high-altitude cloudiness (<440 hPa). The anomaly was not the absolute value but the change of cloudiness in % of the 1X-NOICE value. We changed the figure to be clearer and stay coherent with other anomaly figures, and plotted the absolute change.

Figure 6: Caption: "4xNI ... SOLAR-SOLAR" typo experiment name.

→ Modified.

Figure 9b+d: Instead of shifting the curves to the same equatorial temperatures, it might be better to plot the anomalies due to the individual boundary condition changes (_CO2, ...).

→ This could be an alternative, yes. However, we wanted to keep the shifted curve, as it is very visual to represent what is the 'gradient flattening", what would not be the

case with anomalies.

Figure 10: The red dashed regression line is not visible in my copy.

→ We forgot to remove this from the figure legend. We finally didn't plot the regression line, not to overload the figure. The regression line is visible in the supplementary data file.

Figure 11: Labels are too small.

→ Modified.

3 Language / typos

There are numerous typos and little grammatical errors, as well as some unclear sentences; I would hence recommend copy editing. For example: Page 3, line 85: ...latest "work" are divided... "studies" instead? Page 4, line 96: ...a set of simulation run... (missing plural s) Page 9, lines 208-209: The whole surface is warmer... and which is generally larger over continents... (sentence structure) Page 10, line 251: ...data ant temperature... (and)

→ Corrections done in the revised manuscript

References

Barron, E. J., Fawcett, P. J., Peterson, W. H., Pollard, D. and Thompson, S. L.: A " simulation " of mid-Cretaceous climate Abstract . A series of general circulation model experiments W increased from present day ). By combining all three major variables levels of CO2 . Four times present-day • s W provided the best match to the this, , 10(5), 953–962, 1995. Berner, R. A.: GEOCARBSULF: A combined model for Phanerozoic atmospheric O2 and CO2, Geochim. Cosmochim. Acta, 70(23 SPEC. ISS.), 5653–5664, doi:10.1016/j.gca.2005.11.032, 2006. Bice, K. L., Birgel, D., Meyers, P. A., Dahl, K. A., Hinrichs, K. U. and Norris, R. D.: A multiple proxy and model study of Cretaceous upper ocean temperatures and atmospheric CO2 concentrations,

Paleoceanography, 21(2), 1–17, doi:10.1029/2005PA001203, 2006. Brady, E. C., Deconto, R. M. and Thompson, S. L.: Deep Water Formation and Poleward Ocean Heat Transport in the Warm Climate Extreme of the Cretaceous ( 80 Ma ) evidence, , 25(22), 4205–4208, 1998. Deconto, R. M., Brady, E. C., Bergengren, J. and Hay, W. W.: Late Cretaceous climate, vegetation, and ocean interactions, Warm Clim. Earth Hist., 275–296, doi:10.1017/cbo9780511564512.010, 2000. Goldner, A., Herold, N. and Huber, M.: Antarctic glaciation caused ocean circulation changes at the Eocene-Oligocene transition, Nature, 511(7511), 574–577, doi:10.1038/nature13597, 2014. Hay, W. W., DeConto, R. M., de Boer, P., Flögel, S., Song, Y. and Stepashko, A.: Possible solutions to several enigmas of Cretaceous climate, Springer Berlin Heidelberg., 2019. Hotinski, R. M. and Toggweiler, J. R.: Impact of a Tethyan circumglobal passage on ocean heat transport and "equable" climates, Paleoceanography, 18(1), n/a-n/a, doi:10.1029/2001PA000730, 2003. Kennedy-asser, A. T., Lunt, D. J., Valdes, P. J., Ladant, J., Frieling, J. and Lauretano, V.: Changes in the high latitude Southern Hemisphere through the Eocene-Oligocene Transitionâǎŕ: a model-data comparison, Clim. Past, (September), 1–26, doi:https://doi.org/10.5194/cp-2019-112, 2019. Kennedy, A. T., Farnsworth, A., Lunt, D. J., Lear, C. H. and Markwick, P. J.: Atmospheric and oceanic impacts of Antarctic glaciation across the Eocene-Oligocene transition, Philos. Trans. R. Soc. A Math. Phys. Eng. Sci., 373(2054), doi:10.1098/rsta.2014.0419, 2015. Knorr, G. and Lohmann, G.: Climate warming during antarctic ice sheet expansion at the middle miocene transition, Nat. Geosci., 7(5), 376–381, doi:10.1038/ngeo2119, 2014. Krinner, G., Viovy, N., de Noblet-Ducoudré, N., Ogée, J., Polcher, J., Friedlingstein, P., Ciais, P., Sitch, S. and Prentice, I. C.: A dynamic global vegetation model for studies of the coupled atmosphere-biosphere system, Global Biogeochem. Cycles, 19(1), 1–33, doi:10.1029/2003GB002199, 2005. Ladant, J. B. and Donnadieu, Y.: Palaeogeographic regulation of glacial events during the Cretaceous supergreenhouse, Nat. Commun., 7(April 2017), 1–9, doi:10.1038/ncomms12771, 2016. Lunt, D. J., Haywood, A. M., Schmidt, G. A., Salzmann, U., Valdes, P. J., Dowsett, H. J. and Loptson, C. A.: On the causes of mid-Pliocene warmth and polar amplification, Earth

Planet. Sci. Lett., 321–322, 128–138, doi:10.1016/j.epsl.2011.12.042, 2012. Lunt, D. J., Huber, M., Anagnostou, E., Baatsen, M. L. J., Caballero, R., DeConto, R., Dijkstra, H. A., Donnadieu, Y., Evans, D., Feng, R., Foster, G. L., Gasson, E., Von Der Heydt, A. S., Hollis, C. J., Inglis, G. N., Jones, S. M., Kiehl, J., Turner, S. K., Korty, R. L., Kozdon, R., Krishnan, S., Ladant, J. B., Langebroek, P., Lear, C. H., LeGrande, A. N., Littler, K., Markwick, P., Otto-Bliesner, B., Pearson, P., Poulsen, C. J., Salzmann, U., Shields, C., Snell, K., Stärz, M., Super, J., Tabor, C., Tierney, J. E., Tourte, G. J. L., Tripati, A., Upchurch, G. R., Wade, B. S., Wing, S. L., Winguth, A. M. E., Wright, N. M., Zachos, J. C. and Zeebe, R. E.: The DeepMIP contribution to PMIP4: Experimental design for model simulations of the EECO, PETM, and pre-PETM (version 1.0), Geosci. Model Dev., 10(2), 889–901, doi:10.5194/gmd-10-889-2017, 2017. Maffre, P., Ladant, J. B., Donnadieu, Y., Sepulchre, P. and Goddéris, Y.: The influence of orography on modern ocean circulation, Clim. Dyn., 50(3–4), 1277–1289, doi:10.1007/s00382-017-3683-0, 2018. Monteiro, F. M., Pancost, R. D., Ridgwell, A. and Donnadieu, Y.: Nutrients as the dominant control on the spread of anoxia and euxinia across the Cenomanian-Turonian oceanic anoxic event (OAE2): Model-data comparison, Paleoceanography, 27(4), 1–17, doi:10.1029/2012PA002351, 2012. Otto-bliesner, B. L. and Upchurch, G. R.: the Late Cretaceous period, , 385127(February), 18–21, 1997. Poulsen, C. J., Barron, E. J., Arthur, M. A. and Peterson, W. H.: Response of the mid-Cretaceous global oceanic circulation to tectonic and $CO_2$ forcings, Paleoceanography, 16(6), 576–592, doi:10.1029/2000PA000579, 2001. Poulsen, C. J., Pollard, D. and White, T. S.: General circulation model simulation of the $\delta$18O content of continental precipitation in the middle Cretaceous: A model-proxy comparison, Geology, 35(3), 199–202, doi:10.1130/G23343A.1, 2007. Schmidt, G. A. and Mysak, L. A.: Can increased poleward oceanic heat flux explain the warm Cretaceous climate?, Paleoceanography, 11(5), 579–593, doi:10.1029/96PA01851, 1996. Sepulchre, P., Caubel, A., Ladant, J., Bopp, L., Boucher, O., Braconnot, P., Brockmann, P., Cozic, A., Donnadieu, Y., Estella-perez, V., Ethé, C., Fluteau, F., Foujols, M., Gastineau, G., Ghattas, J., Hauglustaine, D., Hourdin, F., Kageyama, M., Khodri, M., Marti, O., Meurdesoif, Y.,

Mignot, J., Sarr, A., Servonnat, J., Swingedouw, D., Szopa, S. and Tardif, D.: IPSL-CM5A2 . An Earth System Model designed for multi-millennial climate simulations, , (December), 2019. Tabor, C. R., Poulsen, C. J., Lunt, D. J., Rosenbloom, N. A., Otto-Bliesner, B. L., Markwick, P. J., Brady, E. C., Farnsworth, A. and Feng, R.: The cause of Late Cretaceous cooling: A multimodel-proxy comparison, Geology, 44(11), 963–966, doi:10.1130/G38363.1, 2016. Upchurch, G. R., Kiehl, J., Shields, C., Scherer, J. and Scotese, C.: Latitudinal temperature gradients and high-latitude temperatures during the latest Cretaceous: Congruence of geologic data and climate models, Geology, 43(8), 683–686, doi:10.1130/G36802.1, 2015.

[Figure]

*Geopotential height (m) at 850 hPa for piControl and 1X-NOICE simulations and corresponding anomaly.*

[Figure]

**Fig. 1.**

---

## Author Comment (AC2) · 8 Apr 2020

1 General comments

Why should the Cenomanian-Turonian thermal maximum be of more relevance than that of the PETM, MECO, etc. to the future. Worthwhile to flesh out why this particular time period is deemed important.

→ We did not mean to imply that the Cenomanian-Turonian thermal maximum was of more relevance to the future than other greenhouse intervals of the deep-time past. However, the CT thermal maximum can be of particular interest (as can be the PETM)

in that it is a time interval with elevated atmospheric CO2 levels, probably similar or even higher than those expected in the future under a business-as-usual carbon emission scenario. In this manuscript, we present a CT simulation, which has been performed as part of a greater project on OAE2, but we note that Early Eocene and PETM simulations have also been performed by the same group and are part of the DeepMIP project (Lunt et al., Clim. Past 2020, in review). We have clarified the text as follows: "The Cretaceous period is of particular interest to understand drivers of past greenhouse climates because intervals of prolonged global warmth (O'Brien et al. 2017, Huber et al. 2018) and elevated atmospheric CO2 levels (Wang et al., 2014), possibly similar to future levels, have been documented in the proxy record. The thermal maximum of the Cenomanian-Turonian (CT) interval (94 Ma) represents the acme of Cretaceous warmth, during which occurred one of the most important carbon cycle perturbation of the Phanerozoic, the oceanic anoxic event 2 (OAE2). Valuable understanding can hence be drawn from investigations of the mechanisms responsible for the CT thermal maximum and carbon cycle perturbation."

There will also be non-linear feedbacks that cannot be deconvolved with the current methodology. I.e. The proportion of landmass at different latitudes between the pre-industrial and Cretaceous at Preindustrial CO2 levels (and topographic height differences) will have a dissimilar impact on the vegetation response and in turn the warming response from increased CO2.

→ We agree and note that referee #1 also raised this point. We agree that repeating the experiments with a different sequence of changes would be desirable to fully assess the shortcomings of the linear factorization method. It indeed has been suggested that a different sequence of boundary condition changes would probably give different results (Lunt et al., 2012). However, these additional experiments using the IPSL-CM5A2 earth system model require a computational cost that we cannot afford. We have added the following discussion about linear factorization in the revised manuscript:

"The choice of applying a linear factorization approach was made for problems of computing time and cost. Changing the sequence of changes or applying a symmetrical factorization as in Lunt et al (2012) would require too many supplementary simulations. However, such a method is very dependent on the sequence of changes, and results would be probably different if boundary conditions were change in a different order (Lunt et al., 2012)".

I did sometimes get a bit confused with the methodology section. It was sometimes hard to see what models were initialised with what initial conditions. How were some of the boundary conditions treated. i.e. what were the actual albedo prescribed? Was the vegetation uniformly applied even to mountain regions where it may not grow? Might be worth being a bit more concise in describing them individual boundary conditions and how they were implemented for clarity and reproducibility.

→ The description of boundary conditions will be reorganized to be clearer.

2 Minor points.

Page 5 – Line 40. It looks like only the 1X-NOICE is in equilibrium in the deep ocean. The other four still appear to be trending. I think for clarity this should be stated or changed to "near-surface equilibrium". I certainly sympathize as some of my own simulations can take up to 10,000 model years to reach equilibrium. However, I do not think this will change the overall results, but for clarity it should not be stated as complete equilibrium. Gregory plots may also be another useful diagnostic to see if you have an energy imbalance otherwise. → Simulations are indeed not perfectly equilibrated and we have clarified the text: Page 5, line 40: "The piControl simulation was run for 1800 years and the five others for 2000 years in order to reach a near-surface equilibrium (Fig.1)".

Ice sheet removal impact. I agree with your assessment of the regional impact, however it might also be that you get a pseudo ice sheet in the 1X-NOICE simulation with perennial snow cover over the soil surface, just a low elevation one. I suspect this is the case as in the 4X-NOICE you get a much large response in the change in surface

and planetary albedo. Did you ever run a 4xCO2 experiment with ice sheets in the pre-industrial to see the relative impact of just the CO2? → Unfortunately, we did not perform a 4X CO2 simulation with prescribed preindustrial ice sheets. Indeed, we have a snow cover in the 1X-NOICE, but it is also the case in the 4X-NOICE, and the snow cover is even larger over the Antarctic in the 4X-NOICE than in the 1X-NOICE because of a larger amount of precipitations. The large decrease of surface albedo seen between the 1X-NOICE and the 4X-NOICE is due to the decrease of sea ice. We will add figures of surface albedo in the supplementary figures.

See Figures 1, 2, 3 of the response.

Do you see any change in ocean circulation patterns from removal of the ice sheet and increased CO2 in the preindustrial?

→ The global meridional stream function keeps the same structure when removing ice sheets or increasing CO2 in the preindustrial. However, the intensity of ocean circulation changes. In the simulation without ice sheets, NADW is slightly weaker whereas AABW is slightly stronger relative to the preindustrial. The 4x CO2 simulation predicts stronger AABW and NADW with a shallowing of the NADW.

See Figure 4 of the response.

We can see also some local changes of surface circulation around the Antarctic when removing the polar ice cap, that seems to be related to changes in winds. These changes are probably driving the cooling patch that we can see on fig. 4b, that corresponds to a local increase of sea ice. We didn't want to detail this in the manuscript as it is regional changes.

See Figure 5 of the response.

3 Minor comments.

Page 1 & 2 – Line 17 & 33. 'period' not "era"

→ Modified.

I think it is worthwhile to define what you mean by 'high' and 'low' latitudes as you often see different values purported in different studies for clarity.

→ Done: "high-latitudes (> 60° of latitude)"; "low-latitudes (< 30° of latitude)"

Throughout – put references in chronological order. Page 2, Line 60. Define "P.A.L." here. This is the first instance in the ms rather than on page 6- line 50. Page 3 – Line 79. Change sentence to "the primary driver of Cretaceous climate has been suggested". Page 3 – Line 83. Delete erroneous "s".

→ Done.

Page 4 – Line 96. Probably more accurate to say "We performed six simulations using both Pre industrial and Late Cretaceous boundary conditions where we incrementally modify the Pre-industrial boundary conditions to that of the Late Cretaceous for: : :: : :1,2,3,4".

→ Thank you for this suggestion. The sentence was rephrased as: "We performed six simulations, using both preindustrial and Late Cretaceous boundary conditions where we incrementally modify the preindustrial boundary conditions to that of the Late Cretaceous for the following. . .".

Page 4. Although stated that IPSL-CM5A2 has been used for contemporary and future climate simulations it would be worth adding a line that states how well the model performs in simulating a modern-day climate. → The performances of the IPSL-CM5A2 are fully described in (Sepulchre et al., 2019), we added the reference to the manuscript: "Building on technical developments, IPSL-CM5A2 provides enhanced computing performances compared to IPSL-CM5A-LR, allowing thousand-years long integrations required for deep-time paleoclimate applications or long-term future projections (Sepulchre et al., 2019). IPSL-CM5A2 reasonably simulates modern-day and historical climates (despite some biases in the tropics), whose complete description

and evaluation can be found in Sepulchre et al., 2019."

Page 5 – Line 20. Repetition of "long" in the sentence. Remove one of them. Page 5 – Line 37. "retreat" to retreat. But I think 'removed' would be more accurate. → Done.

Page 6 – Line 53. Is that from the 4X-NOICE simulation? → We will give more precisions in the manuscript (see also reponse to comment from reviewer #1): we have adapted the Lunt et al., 2017 formulation has the Cenomanian-Turonian ocean was warmer (see also Sepulchre et al., 2019) :

If depth $\leq$ 1000 m :

T=10+((1000-depth)/1000)*25 cosâĄą(latitude)

if depth > 1000 m :

T = 10

Page 6 – Line 218-222. Did you look at atmospheric stability arguments in relation to this?

→ Indeed, we should look at atmospheric stability to explain the observations made in relation to these processes that are quite complex. We would need to go into details regarding the changes in water content, relative/specific/absolute humidity as well as atmospheric dynamics to explain the observed changes. We do not want to go into such details in the manuscript which is already long and which not specifically focus on the impact of a $CO_2$ increase, so we finally decided to remove this paragraph.

Page 6 – Line 223. Do you mean greater season ice melt or there being less sea ice area in the 4X-NOICE compared to the 1X-NOICE simulation?

→ Less sea ice in the 4X-NOICE: "The sea level pressure decrease is possibly a feedback driven by reduced sea ice cover and associated higher temperatures."

Page 12 – Section 3.5.1. The percentage change adds to 99%. Rounding error?

→ The 30% was actually a 31%, but we finally decided to remove the percentages as it was confusing and to keep only raw values (see also comments from reviewer #1).

Page 12 –Line 321-324: There does not appear to be any change in the N.Hem SST gradients (0.45/lat). Any idea why? You attribute the Greenland ice sheet/sea ice for less sensitivity in the atmospheric gradient of the N.Hem. Something similar here?

→ Indeed, by calculating the gradient with the temperature value at 80°N it seems that there is no flattening. However, if we look at the figure 9b, we can see that the gradient is flatter until 70° of latitude North and is very steep beyond. It light be because the arctic ocean is very isolated due to the paleogeographic configuration, which doesn't allow heat transport beyond 80°N.

Page 14 – Line 370. Do you mean that you used the Tabor, et al. dataset and adding more data points

→ Yes, we clarified it : "Our SST data compilation is modified from Tabor et al (2016), with additional data from more recent studies (see Supplementary data)."

Page 14 –Line 388. Only if you suggest there is a seasonal proxy bias. This is mentioned later on, but might be worth a few ref's that show there are seasonal bias in some proxies.

→ Done: "This congruence would imply that a seasonal bias may exist in temperatures reconstructed from proxies, which is suggested in previous studies (Sluijs et al., 2006; Hollis et al., 2012; Huber, 2012)."

Page 16 – Line 448. Do you mean 'more complex' rather than "large"?

→ Modified sentences: "The signal is notably due to a 9°C warming in response to the fourfold increase in pCO2, which converts to an increase of 4.5°C for a doubling of pCO2 (assuming that the response is linear). This sensitivity agrees with the higher end of the range of values in the investigations mentioned above. However, the sensitivity of IPSL-CM5A2 in our simulations could be slightly lower as the simulations are not

completely equilibrated – see Figure 1)."

Page 17 – Line 480. "cloud" not "clouds".

→ Modified.

Page 17 – Conclusions section. This is a tad bit repetitive of the results/discussion. Perhaps broaden this out with respect to the discussion in your introduction.

→ We will arrange the conclusions to be less repetitive and to fit with raised questions in the introduction

Figure 1, 5,6, 8, 9, 11 Captions. Add 'mean annual' to caption.

→ Done.

Figure 2. Is that the model resolution geographies?

→ Bathymetry was shown at the model resolution but not the topography that is shown with a bilinear interpolation. We will change the figure to show the model resolution for the topography also.

Figure 11b. Appears to be some modelling studies missing? E.g. HadCM3L 2xCO2 data point and others. Or was I interpreting this wrongly? Quite possible!

→ Indeed, some points were missing! We added them.

References

Hollis, C. J., Taylor, K. W. R., Handley, L., Pancost, R. D., Huber, M., Creech, J. B., Hines, B. R., Crouch, E. M., Morgans, H. E. G., Crampton, J. S., Gibbs, S., Pearson, P. N. and Zachos, J. C.: Early Paleogene temperature history of the Southwest Pacific Oceanâ Ăř: Reconciling proxies and models, Earth Planet. Sci. Lett., 349–350, 53–66, doi:10.1016/j.epsl.2012.06.024, 2012. Huber, M.: Progress in Greenhouse Climate Modeling, Paleontol. Soc. Pap., 18, 213–262, doi:10.1017/s108933260000262x, 2012. Lunt, D. J., Haywood, A. M., Schmidt, G. A., Salzmann, U., Valdes, P. J., Dowsett,

H. J. and Loptson, C. A.: On the causes of mid-Pliocene warmth and polar amplification, Earth Planet. Sci. Lett., 321–322, 128–138, doi:10.1016/j.epsl.2011.12.042, 2012. Lunt, D. J., Huber, M., Anagnostou, E., Baatsen, M. L. J., Caballero, R., De-Conto, R., Dijkstra, H. A., Donnadieu, Y., Evans, D., Feng, R., Foster, G. L., Gasson, E., Von Der Heydt, A. S., Hollis, C. J., Inglis, G. N., Jones, S. M., Kiehl, J., Turner, S. K., Korty, R. L., Kozdon, R., Krishnan, S., Ladant, J. B., Langebroek, P., Lear, C. H., LeGrande, A. N., Littler, K., Markwick, P., Otto-Bliesner, B., Pearson, P., Poulsen, C. J., Salzmann, U., Shields, C., Snell, K., Stärz, M., Super, J., Tabor, C., Tierney, J. E., Tourte, G. J. L., Tripati, A., Upchurch, G. R., Wade, B. S., Wing, S. L., Winguth, A. M. E., Wright, N. M., Zachos, J. C. and Zeebe, R. E.: The DeepMIP contribution to PMIP4: Experimental design for model simulations of the EECO, PETM, and pre-PETM (version 1.0), Geosci. Model Dev., 10(2), 889–901, doi:10.5194/gmd-10-889-2017, 2017. Sepulchre, P., Caubel, A., Ladant, J., Bopp, L., Boucher, O., Braconnot, P., Brockmann, P., Cozic, A., Donnadieu, Y., Estella-perez, V., Ethé, C., Fluteau, F., Foujols, M., Gastineau, G., Ghattas, J., Hauglustaine, D., Hourdin, F., Kageyama, M., Khodri, M., Marti, O., Meurdesoif, Y., Mignot, J., Sarr, A., Servonnat, J., Swingedouw, D., Szopa, S. and Tardif, D.: IPSL-CM5A2 . An Earth System Model designed for multi-millennial climate simulations, , (December), 2019. Sluijs, A., Schouten, S., Pagani, M., Woltering, M. and Brinkhuis, H.: Subtropical Arctic Ocean temperatures during the Palaeocene / Eocene thermal maximum, , (May 2014), doi:10.1038/nature04668, 2006. Tabor, C. R., Poulsen, C. J., Lunt, D. J., Rosenbloom, N. A., Otto-Bliesner, B. L., Markwick, P. J., Brady, E. C., Farnsworth, A. and Feng, R.: The cause of Late Cretaceous cooling: A multimodel-proxy comparison, Geology, 44(11), 963–966, doi:10.1130/G38363.1, 2016. Wang, Y., Huang, C., Sun, B., Quan, C., Wu, J. and Lin, Z.: Paleo-CO2 variation trends and the Cretaceous greenhouse climate, Earth-Science Rev., 129, 136–147, doi:10.1016/j.earscirev.2013.11.001, 2014.
* * *
[Figure]

[Figure]

Snow mass annual average (kg/m2) over the Antarctic for 1X-NOICE and 4X-NOICE simulations and corresponding anomaly.

**Fig. 1.**

[Figure]

Surface albedo average (%) in the Austral ocean for 1X-NOICE and 4X-NOICE simulations and corresponding anomaly.

[Figure]

Surface albedo annual average (%) in the Arctic for 1X-NOICE and 4X-NOICE simulations and corresponding anomaly.

**Fig. 2.**

[Figure]

Sea-ice annual average (%) in the Austral ocean for 1X-NOICE and 4X-NOICE simulations and corresponding anomaly.

[Figure]

Sea-ice annual average (%) in the Arctic for 1X-NOICE and 4X-NOICE simulations and corresponding anomaly.

**Fig. 3.**

[Figure]

*Global meridional stream function (Sv) for piControl, 1X-NOICE and 4X-NOICE simulations. Positive values (red) indicate a clockwise circulation and negative values (blue) an anticlockwise circulation.*

**Fig. 4.**

[Figure]

[Figure]

*Currents velocity for piControl (left) and 1X-NOICE (right) simulations. The red arrow shows the ACC position. The blue box shows the position of the sea ice increase (=cooling area).*

[Figure]

*Sea surface wind speed anomaly during winter (1X-NOICE-piControl). The polar ice cap removal causes a displacement of winds and thus ACC to the North, which allows an increase of sea ice in the 1X-NOICE simulation. Blue box: Sea ice anomaly, Black arrow: ACC position for the piControl simulation, Green arrow: ACC position for the 1X-NOICE simulation (after polar cap removal).*

**Fig. 5.**